


# Simulation study for the Stratospheric Inferred Winds (SIW) sub-millimeter limb sounder

Philippe Baron[1], Donal Murtagh[2], Patrick Eriksson[2], Jana Mendrok[2,3], Satoshi Ochiai[1], Kristell Pérot[2], Hideo Sagawa[4], and Makoto Suzuki[5]

[1]National Institute of Information and Communications Technology, 4-2-1 Nukui-kitamachi, Koganei, Tokyo 184-8795, Japan
[2]Department of Space, Earth and Environment, Chalmers University of Technology, 41296 Göteborg, Sweden
[3]At Chalmers University of Technology until Dec. 2017
[4]Kyoto Sangyo University, Kyoto, Japan
[5]Japan Aerospace Exploration Agency, Tsukuba, 305-8505 Japan

*Correspondence to:* P. Baron (baron@nict.go.jp)

**Abstract.** Stratospheric Inferred Winds (SIW) is a Swedish mini sub-millimeter limb sounder selected for the 2[nd] InnoSat platform launch planned near 2022. It is intended to fill the altitude gap between 30–70 km in atmospheric wind measurements and also aims at pursuing the limb observations of temperature and key atmospheric constituents between 10–90 km when current satellite missions are probably stopped. Line-of-sight winds are retrieved from the Doppler shift of molecular emission lines introduced by the wind field. Observations will be performed with two antennas pointing toward the limb with perpendicular directions to reconstruct the 2-D horizontal wind vector. Each antenna has a vertical field of view of 5 km. The chosen spectral band near 655 GHz contains a dense group of strong $O_3$ lines suitable for exploiting the small wind information in stratospheric spectra. Using both sidebands of the heterodyne receiver, a large number of chemical species will be measured including $O_3$-isopotologues, $H_2O$, HDO, HCl, ClO, $N_2O$, $HNO_3$, NO, $NO_2$, HCN, $CH_3CN$ and $HO_2$. This paper presents the simulation study for assessing the measurement performances. The line-of-sight winds are retrieved between 30–90 km with the best sensitivity between 35–70 km where the precision (1-$\sigma$) is 5–10 $m\,s^{-1}$ for a single scan. Similar performances can be obtained during day and night conditions except in the lower mesosphere where the photo-dissociation of $O_3$ in daytime reduces the sensitivity by 50 % near 70 km. Profiles of $O_3$, $H_2O$ and temperature are retrieved with a high precision up to 50 km ($< 1$ %, $< 2$ %, 1 K, respectively). Systematic errors due to uncertainties on spectroscopic parameters, on the radiometer sideband ratio and in the radiance calibration process are investigated. A large wind retrieval bias of 10–30 $m\,s^{-1}$ between 30–40 km can be induced by the air-broadening parameters uncertainties of $O_3$ lines. This highlights the need for a good knowledge of these parameters and to study methods to mitigate the retrieval bias.

## 1 Introduction

Millimeter and sub-millimeter (MM and SMM) limb sounders have been successfully used for more than two decades to probe the atmospheric composition and the temperature from the upper-troposphere to the lower thermosphere (Waters et al., 1993; Murtagh et al., 2002; Waters et al., 2006; Kikuchi et al., 2010). The first generation of Millimeter Limb Sounder (MLS)



provided unique observations of ClO, $O_3$, $H_2O$ and $HNO_3$ allowing, for instance, a better understanding of the physical and chemical processes leading to the northern high-latitude $O_3$ depletion (Waters et al., 1993). Subsequent SMM missions have allowed the monitoring of the middle-atmosphere (15–110 km) almost without interruption since the first MLS and have significantly contributed to the current middle-atmospheric measurement database (Hegglin and Tegtmeier, 2017). However,

no successors of these missions are planned yet, and there is a risk of an observation gap in the near future.

The advantages of such observations are manifold. The thermal emission spectrum at MM and SMM wavelengths is rich of isolated spectral lines from asymmetric molecules and molecular oxygen. Some of these lines are the clearest signal within the whole atmospheric spectrum of important chemical species such as $HO_2$ and ClO (Urban et al., 2005; Khosravi et al., 2013; Sagawa et al., 2013; Millán et al., 2015). The $O_2$ lines give temperature and pressure, and the limb geometry provides

a suitable vertical resolution for describing the middle-atmosphere. Molecules are sensed in the thermal equilibrium state with no diurnal difference in the measurement performance, and the measurement is not perturbed by stratospheric polar-clouds and aerosols. Furthermore, the technology is mature allowing missions to operate over a period longer than a decade. Methods have already been used for improving the horizontal resolution with tomographic observations (Livesey et al., 2006; Christensen et al., 2015) or for obtaining very high signal-to-noise ratio using 4-K cryogenic cooling (Kikuchi et al., 2010).

Modeling middle-atmospheric major dynamical phenomena such as high-latitude sudden-stratospheric warming or equatorial quasi-biennial oscillation are still challenging (Limpasuvan et al., 2012; Newman et al., 2016; Orsolini et al., 2017). Wind is one of the primary parameters for describing the physical state of the atmosphere but models have difficulties to reproduce it where the atmospheric flow cannot be described by the geostrophic approximation, such as in the equatorial region where the Coriolis force is weak and, in the upper stratosphere and mesosphere where wave and tides phenomena tend to dominate the

wind fields (Baron et al., 2013b; Le Pichon et al., 2015; Duruisseau et al., 2017; Rüfenacht et al., 2017). The middle atmosphere becomes more critical since climate and weather models expand into the mesosphere for improving their long term prediction capability (Baldwin et al., 2003; Hoppel et al., 2008; Baldwin et al., 2010; Gerber et al., 2012). Though there is a strong need for middle-atmospheric wind measurements to validate and constrain the models, only high altitude (>90 km) wind measurements with optical sensors currently exist on a global scale (Shepherd, 2015). Ground-based stations does not cover the globe

uniformly and most of the data are limited to heights below 30 km (Ishii et al., 2017) or above 70 km (Baumgarten, 2010). However recent efforts have been made to close this altitude gap (Rüfenacht et al., 2014; Le Pichon et al., 2015; Blanc et al., 2018).

Providing wind data in the middle atmosphere from space is one of the challenges for future missions. The European Space Agency is going to launch this year the Atmospheric Dynamics Mission equipped with a wind lidar to demonstrate the

feasibility of such measurements (Stoffelen et al., 2005). However a lidar is well suited for measuring wind in the troposphere but has poor precision above 20–30 km (Ishii et al., 2017). The Stratospheric Wind Interferometer For Transport studies (SWIFT) has been studied by the Canadian Space Agency for deriving winds between 15–45 km from $O_3$ infra-red emission lines (Rahnama et al., 2013). The mission was originally planned for 2010 but it is now very uncertain. Using measurements from the second MLS and from the Superconducting Submillimeter-Wave Limb-Emission Sounder (SMILES), it has been

demonstrated that line-of-sight wind can be retrieved between 30–90 km from the small Doppler shift of MM and SMM





lines (Wu et al., 2008; Baron et al., 2013b). Wind is one of the main outcomes of SMILES-2 that is proposed to the Japan Aerospace Exploration Agency (JAXA) (Ochiai et al., 2017). It is a large instrument (>500 kg) using cryogenic SMM and THz receivers designed for very high sensitive observations between 15–150 km but with a late launch near 2025. Two smaller missions are studied with the possibility to be launched as soon as 2020–22. Wu et al. (2016) propose a small instrument for

measuring the atomic oxygen line at 2.06 THz in order to retrieve its abundance as well as temperature and wind in the lower thermosphere. However this mission cannot provide stratospheric and lower mesospheric information. The second proposal is the Stratospheric Inferred Winds (SIW). It is a small and low-cost satellite mission studied within the Swedish Innosat program (Lindberg, 2016). Through this program, it is planned to launch a scientific mission every two years, and SIW has been selected for the 2nd launch near 2022. SIW can provide atmospheric data between 15–90 km, including the horizontal-

wind vector within 30–90 km. The other primary products are the profiles of temperature, $O_3$, $H_2O$ and more than a dozen of other chemical species. With this mission it will be possible to ensure the continuous monitoring of the middle-atmosphere avoiding a SMM measurement gap.

In this paper we present a simulation study to assess the potential of SIW. A special focus is put on the main parameters: wind, temperature, $O_3$ and $H_2O$ that are derived from the strongest lines in the selected spectral bands. Section 2 describes the

15 mission and the observation technique. The measurement simulation and the retrieval method are explained in Sect. 3 and 4, respectively. The measurement performances are discussed in Sect. 5 and concluding remarks are given in the final section.

## 2 Mission description

### 2.1 Observation and instrument characteristics

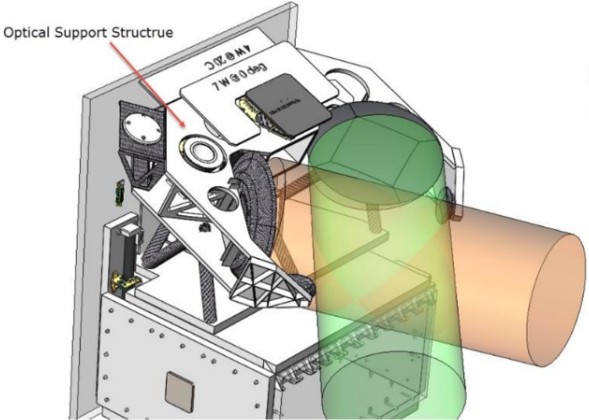

**Figure 1.** View of the Innosat satellite. The box in the lower-part is the platform service module. Above it is the scientific payload including the two antennas. The field of views are represented with green and beige colors (from Omnisys instruments co).



The scientific payload (Fig. 1) and observations characteristics are summarized in Table 1. This is the proposed setting which can still be slightly modified. The platform will be set on a sun-synchronous polar orbit at an altitude of 550 km. It will fly near the terminator crossing the equatorial ascending node at 18:00 local-time (LT). Atmospheric observations will be performed toward the night side using two antennas looking perpendicularly to each other with angles from the satellite velocity of 45°

and 135°, respectively. The antennas will point toward close air-masses with few minutes delay (Fig. 2). They are fixed on the platform and the whole satellite will nod up and down in order to scan the limb alternatively upward and downward from about 15 to 90 km. Using the line-of-sight wind retrieved with the two antennas over close regions will allow us to derive the meridional and zonal wind components (Appendix A).

The signals from the antennas are alternatively sent to a single radiometer passively cooled to 70 K below the ambient

temperature, and analysed with an auto-correlator spectrometer. The forward antenna is used during the upward scans and the aftward one during the downward scans. The heterodyne radiometer operates in double-sideband (DSB) mode yielding to the superposition in the measured spectrum of the two image bands with respect to the local oscillator. The bandwidth and resolution are 8 GHz and 1 MHz, respectively.

The strategy for acquiring the calibration data is not definitively decided yet and will probably be optimized in the future.

Currently the plan is as follows. A calibration load onboard the plaform (black body at ambient temperature) is viewed at the bottom and top of each scan during the turnaround. While limb scanning, the atmosphere and cold-sky are observed alternatively with an integration time of 0.5 s each. Hence, atmospheric spectra are obtained every 2.3 km with an effective vertical resolution of about 5 km.

## 2.2 Spectral bands

The measured spectrum is composed of molecular lines spectrally resolved (Fig. 3). Using a radiative transfer model, they are inverted to retrieve geophysical information. Volume mixing ratio (VMR) and temperature are retrieved from their amplitude, whereas tangent-height pressure and line-of-sight (LOS) wind are retrieved from the width and the frequency position of the lines, respectively.

The Doppler shift induced by the LOS wind (2 kHz for $1 \, \mathrm{m \, s^{-1}}$) is small compared to the line broadening ($1 - 100$ MHz).

This gives a very weak signal to exploit, especially in the lower stratosphere. Baron et al. (2013a) have shown that the spectral region near 655 GHz is the most suitable one for measuring wind with the current hardware. It contains a dense group of strong $O_3$ lines (second row of Fig. 3), that increases by at least a factor 2 the wind measurement sensitivity between 40– 70 km compared to other spectral regions. This band also allows us to retrieve temperature with a good precision in the stratosphere without measuring an $O_2$ line.

The local-oscillator frequency has been carefully selected in order to include as many as possible spectral lines and to reduce the line superposition from both sidebands. Hence lines of chemical species such as HCN (620.3 GHz), $H^{37}Cl$ (625.0 GHz) $H^{35}Cl$ (625.9 GHz), $^{35}ClO$ (649.5 GHz), NO (651.1 GHz) and $N_2O$ (652.8 GHz) are clearly visible. A strong $H_2O$ line is located at 620.7 GHz but very close to an $O_3$ line with similar strength. Lines from around twenty molecules are available



**Table 1.** Characteristics of the SIW payload and observations. The relationship between tangent-height and LOS angle is derived for an Earth radius of 6370 km and a satellite altitude of 550 km above the geoid.

| | |
|---|---:|
| Payload volume | $40{\times}70{\times}40$ cm$^3$ |
| Payload mass/power | 17 kg/47 W |
| Antenna diameter | 30 cm |
| Satellite altitude | 500–600 km |
| Orbit inclination | 98 ° (sun synchronous) |
| Latitude range | 65 °S–82 °N |
| Local time of ascending node | 18:00 |
| Scan altitude | 10–90 km |
| LOS nadir angle | 67.25–69.03 ° (1.78 °) |
| Scan velocity | 0.05 ° s$^{-1}$ (35 s/scan) |
| Spectrum integration time | 0.5 s (1.14 km$^*$) |
| Antenna vertical FOV | 5 km |
| DSB system temperature | 1000–1200 K |
| ACS Bandwidth | 8 GHz |
| ACS resolution | 1 MHz |
| LO frequency | 638.075 GHz ($\lambda =0.47$ mm) |
| IF frequency | 10.075–18.075 GHz |
| Frequency ⇔ velocity | 1 m s$^{-1}$ ⇔ 2 kHz |

$^*$ Tangent point vertical displacement

though some are very weak such as $H_2CO$, $CH_3Cl$ or BrO. Finally let's note that most of the lines where IF $> 14$ GHz have already been measured with Aura/MLS and JEM/SMILES.

## 3 Measurement modeling

### 3.1 Radiative transfer and instrument

5  The signal is a spectral and spatial average of specific intensities (W m$^{-2}$ sr$^{-1}$ Hz$^{-1}$) over narrow instrumental functions. It is expressed in the so-called brightness temperature $T_b$ equal to (Urban et al., 2004)

$$T_b\,(\theta_j,\,\vartheta_i)\,=\,\kappa_b\!\!\int\limits_{\Delta\vartheta}\!d\vartheta\quad g^{sp}\,(\vartheta-\vartheta_i)\left\{w_{lsb}(\vartheta)\!\!\int\limits_{\Delta\theta}\!G_e^{ant}\,(\theta-\theta_j,\,\nu_{lo}-\vartheta)\,I(\theta,\nu_{lo}-\vartheta)\,d\theta\right.$$
$$\left.+\,(1-w_{lsb}(\vartheta))\!\!\int\limits_{\Delta\theta}\!G_e^{ant}\,(\theta-\theta_j,\,\nu_{lo+\vartheta})\,I(\theta,\nu_{lo}+\vartheta)\,d\theta\right\},$$

(1)





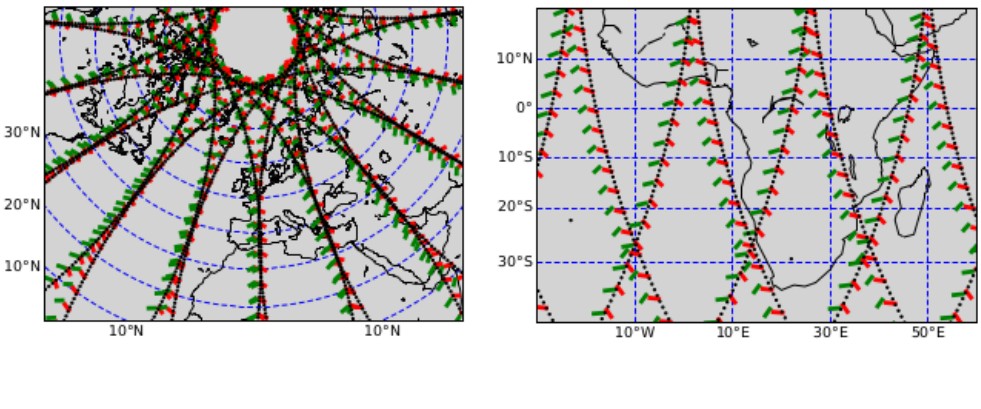

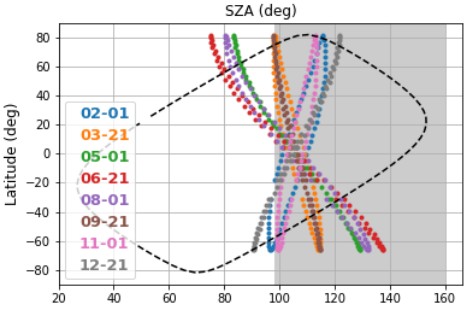

**Figure 2.** The upper panels show the footprints of the forward ($45°$) and aftward ($135°$) views over a 24-hour period. the forward antenna is used during the upward scans (red lines) and the aftward one during the downward scans (green lines). The first tangent point of the upward-scans are located on the black-dotted lines. The lower panel shows the solar zenith angles with respect to latitudes for various days representative of the seasonal variation (colored dots) together with those of the AURA/MLS data (DJF, 2011) used in the simulations (dashed line). The shaded area shows the nighttime measurements in the mesosphere where the $O_3$ diurnal variation is the strongest.

where $i$ is the frequency of the $i$th spectral component of the measurement, $\theta_j$ is the mean nadir angle during the measurement integration time of the $j$th spectra of the scan, and $I$ is the specific intensity which is associated with a LOS such as that shown in Fig. 4 (see Sect. 3.2). The heterodyne receiver is sensitive to atmospheric radiation at frequencies $\nu_{\mathrm{lo}} \pm \vartheta$ where $\nu_{\mathrm{lo}}$ and $\vartheta$ are the local-oscillator and intermediate frequencies (Tab. 1). The instrumental functions are the spectrometer channel response $g_{\mathrm{sp}}$ ($\mathrm{Hz}^{-1}$), the relative weight of the radiometer sidebands $w_{\mathrm{lsb}}$, and the effective antenna pattern $G_{\mathrm{e}}^{\mathrm{ant}}$. The parameter $\kappa_b$ is the Rayleigh-Jeans factor, used to convert the intensity into brightness temperature:

$$\kappa_b = \frac{c^2}{2\,k_b\,\nu_{lo}^2},$$

where $c = 2.997924 \times 10^8 \ \mathrm{m\,s^{-1}}$ is the speed of light in vacuum and $k_{\mathrm{b}} = 1.380662 \times 10^{-23} \ \mathrm{J\,K^{-1}}$ the Boltzmann constant. The spectrometer channel response is assumed to be Gaussian with a Full-Width-Half-Maximum (FHWM) of 1 MHz. The



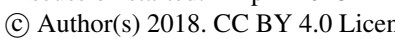

**Figure 3.** Contribution of the most relevant chemical species to the SIW spectrum. More that 20 molecules are shown in 4 groups of two panels. In each group, the upper panel shows the lower sideband spectrum (dashed black lines) with a central frequency of 623.1 GHz and the lower panel shows the upper sideband one with a central frequency of 653.05 GHz. The colored lines are single-molecule spectra. The tangent height is 30 km and frequencies are ordered according to the intermediate frequencies. The intensity is given in brightness temperature (y-axis).





antenna pattern is approximated by a Gaussian function with the FWHM:

$$\sigma_e^{\text{ant}} = \sqrt{\left(\frac{1.22}{D}\frac{c}{\nu_{lo}}\right)^2 + \left(\dot{\theta}\Delta_T\right)^2}, \qquad (2)$$

where $D$ (m) is the antenna diameter, $\dot{\theta}$ ($\text{rad s}^{-1}$) is the vertical scan velocity and $\Delta_T$ is the spectrum integration time. A constant sideband weight is used, $w_{\text{lsb}}(\vartheta) = 0.5$. The integrals in Eq. 1 are computed over ranges $\Delta\theta$ and $\Delta\vartheta$ set to 3 times

5 the FWHM of their corresponding response functions.

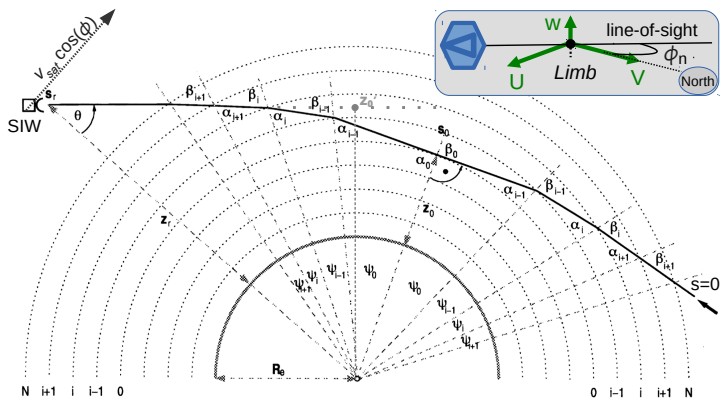

**Figure 4.** Limb sounding geometry for a refracted line-of-sight (full line) and a none refracted one (dashed line). The panel on the right-upper corner shows the orientation of the LOS with respect to the wind components at the tangent point. Figure is adapted from Urban et al. (2004).

### 3.2 Specific intensity and wind

The specific intensity is computed using the radiative transfer equation:

$$I(\theta,\nu) = \int_{s=0}^{s_r} B_\nu(s)\,K_\nu\left(s,\{\nu_a(s)\}_{\text{lines}}\right) \exp\left(-\int_{s'=s}^{s_r} K_\nu\left(s,\{\nu_a(s')\}_{\text{lines}}\right) ds'\right) ds, \qquad (3)$$

where $s$ indicates the position on the LOS, $B$ is the Planck function ($\text{W m}^{-2}\,\text{sr}^{-1}\,\text{Hz}^{-1}$) and $K$ ($\text{m}^{-1}$) is the absorption

coefficient. The background cosmic radiation ($T_b \approx 1$ mK) is neglected. The absorption coefficient is computed with a line-by-line and continua models (Urban et al., 2004). The spectroscopic parameters describing the molecular lines are taken from the HITRAN catalog (Rothman et al., 2009) except those for $\text{BrO}$, $\text{CH}_3\text{Cl}$ and $\text{CH}_3\text{CN}$ that are from the Jet Propulsion Laboratory (Pickett et al., 1998). The frequency of the spectral lines viewed from the receiver ($\{\nu_a(s)\}_{\text{lines}}$) depends on the mean relative motion of the molecules with respect to the receiver, i.e., satellite velocity and wind. The Doppler-shift effect on

the Planck function is neglected. The line frequency is also shifted by the atmospheric pressure but this effect is small above 25 km where winds are measured.





A spherical Earth is assumed for assessing the impacts of all the parameters contributing to the line Doppler shift. At a height $z_i$ and for a LOS nadir angle $\theta$, the line apparent frequency is (Kursinski et al., 1997)

$$\nu_a(\theta, z_i) = \nu_0 \left(1 - \frac{\left[V(z_i)\cos(\phi_n) + (U(z_i) + \omega_e R_e \cos(\Lambda))\sin(\phi_n)\right]\sin(\alpha_i)}{c} + \frac{W\cos(\alpha_i)}{c} + \frac{V_{sat}\cos(\phi)\sin\theta}{c}\right) \quad (4)$$

where $\nu_0$ (Hz) is the rest frequency of the transition, $V_{sat}$ is the satellite velocity with respect to a fixed frame attached to the

Earth center, $(U, V, W)$ is the 3-D wind velocity defined with respect to the Earth surface, and $\omega_e$, $\Lambda$ and $Re$ are the Earth rotation angular velocity $(\text{rad s}^{-1})$, the latitude and the geoid radius at the position $i$. The LOS nadir angle at $z_i$ is $\alpha_i$, and $\phi_n$ is the angle between the LOS and the north direction (Fig. 4). At the tangent height point $(i = 0)$, $\alpha_0 = 90°$ and the Doppler shift $\delta\nu(\theta, z_0)$ is

$$\delta\nu(\theta, z_0) = -\frac{\nu_0}{c}\left(V_{los}(z_0) + \omega_e R_e \cos(\Lambda)\sin(\phi_n) - V_{sat}\cos(\phi)n_0\frac{z_0 + R_e}{z_r + R_e}\right), \quad (5)$$

where $z_r$ is the receiver height, $n_0$ is the refractive index at the tangent point, $\sin(\theta) = n_0\frac{z_0 + R_e}{z_r + R_e}$ and $V_{los}$ is the LOS component of the horizontal wind:

$$V_{los}(z_0) = V(z_0)\cos(\phi_n) + U(z_0)\sin(\phi_n). \quad (6)$$

At the equator and for the forward LOS, the Doppler shifts due to the satellite velocity and to the Earth rotation are $\approx -8\,\text{MHz}$ $(+4000\,\text{m s}^{-1})$ and $\mp 0.74\,\text{MHz}$ $(\pm 370\,\text{m s}^{-1})$, respectively. In order to simplify the calculations, we consider the case of a

pseudo LOS-wind profile which, unlike a real one, induces a Doppler-shift $\delta\nu(z) = -\nu_0/c\,V_{plos}(z)$ that is independent of the angles $\alpha_i$ and Earth rotation, and includes the vertical changes due to the satellite velocity:

$$V_{plos}(z) = V_{los}(z) - V_{sat}\cos(\phi)n_z\left(\frac{z - 50\,\text{km}}{z_r + R_e}\right). \quad (7)$$

At the tangent point, the pseudo wind induced the same Doppler-shift as that given in Eq. (5) to within the same constant over the full vertical scan. The constant includes the Earth rotation effects and most of the satellite velocity ones. The terms

embedded in this constant are known with a precision better than $1\,\text{m s}^{-1}$ using the star-trackers and GPS data onboard the satellite. Such a setting is chosen to yield the satellite-velocity induced Doppler-shift to zero at $z = 50\,\text{km}$, center of the vertical scan.

The pseudo-wind approximation induces errors on the line apparent frequency at positions on the LOS other than the tangent point. These errors are small and have negligible impacts on the retrievals. Indeed, VMR and temperature retrievals are not

sensitive to small frequency errors, and regarding wind retrieval, the information is extracted from optically thin measurements which are characterized by narrow specific-intensity weighting functions peaking at the tangent point.

### 3.3    Calibration and measurement noise

The raw intensity delivered by the spectrometer is expressed as (Olberg et al., 2003):

$$C_{i,j} = G_{i,j}\left[T_{sys}(i,j) + \eta_x T_b(i,j) + (1 - \eta_x)T_{so}(i)\right] \quad (8)$$




with $i$ and $j$ are the tangent height and frequency indices, $T_{sys}$ is the double sideband system temperature, $T_{so}$ is the antenna spill-over, $\eta_x$ is the efficiency of the integrated antenna ($x$=a) or hot-load horn ($x$=c), and $G$ is the radiometric gain. The last is

$$G_{i,j} = g_{i,j}\left(1 - \alpha\langle C_{i,j}\rangle\right) \tag{9}$$

where $\langle\bullet\rangle$ denotes the average over the frequencies $j$ and $\alpha$ is a positive coefficient to account for a non-linear response of the radiometer (Ochiai et al., 2013). The double-sideband system temperature of SIW is expected to be about 1100 K (OMNISYS, private communication). The signal intensity is calibrated using the emissions from the cold sky with a Rayleigh-Jeans temperature $T_c \approx 10^{-3}$ K, and from an ambient temperature hot-load (Rayleigh-Jeans temperature $T_h \approx 250$ K) measured between two scans. A linear response of the radiometer is assumed and its gain is derived as (Olberg et al., 2003)

$$\widehat{G}_j = \frac{\overline{C}_h(j) - \overline{C}_c(j)}{\epsilon T_h}, \tag{10}$$

where $\epsilon$ is the hot-load emissivity, and $C_h$ and $C_c$ are the receiver raw outputs for the hot load and cold-sky. The upper-bars $\overline{\bullet}$ indicate that an average value over the whole scan is used. The brightness temperature of the atmospheric signal is then

$$\widehat{T}_b(i,j) = \frac{C_{atm}(i,j) - \overline{C}'_c(i,j)}{\eta_a \widehat{G}_{i,j}} + \text{offset}_i, \tag{11}$$

where $C_{atm}$ is the receiver output when the atmosphere is viewed and $\overline{C}'_c(i,j)$ is the cold sky output interpolated at the $C_{atm}$ time. We consider that during the scan, the atmosphere and cold-sky are viewed alternatively during 0.5 sec each. The second term of the equation is a tangent-height dependent offset induced by the antenna spillover. Such radiance offset is retrieved together with the geophysical information and it is not considered as a retrieval error source. The brightness temperature error from the radiometer noise and the calibration model is

$$
\begin{aligned}
\delta\widehat{T}_b(i,j) &= \frac{\delta C_{atm}(i,j)}{\widehat{G}_j} + \frac{\delta\overline{C}'_c(i,j)}{\widehat{G}_j} + \widehat{T}_b(i,j)\frac{\delta\widehat{G}_j}{\widehat{G}_j} + \widehat{T}_b(i,j)\frac{\delta\eta_a}{\eta_a} + e_{NL}(i,j) \\
&= \epsilon_{atm}(i,j) + \epsilon'_{\overline{c}}(i,j) + \left(\epsilon_{\overline{h}}(j) + \epsilon_{\overline{c}}(j)\right)\beta_h(i,j) + \left(e_{\overline{h}} + e_{\eta_a}\right)\widehat{T}_b(i,j) + e_{NL}(i,j)
\end{aligned} \tag{12}
$$

where $\epsilon_{atm}$ and $\epsilon'_{\overline{c}}$ are white noises on the atmospheric and cold-sky brightness temperatures (Eq. 11), $\epsilon_{\overline{c}}$ and $\epsilon_{\overline{h}}$ are those on the hot-load and cold-sky spectra in Eq. (10), and $\beta_h = \frac{T_b(i,j)}{\epsilon T_h}$. The two last elements of the equation are systematic errors induced by relative errors $e_{\overline{h}}$ and $e_{\eta_a}$ on the hot-load emission ($\epsilon T_H$) and the antenna efficiency ($\eta_a$), and the error due to the receiver non-linearity ($e_{NL}$).

The noise standard-deviation is given by the radiometric equation (Jarnot et al., 2006; Ochiai et al., 2013):

$$\sigma_t(i,j) = \left[T_{sys}^{dsb}(i,j) + T_b(i,j)\right]\sqrt{\frac{1}{wt} + \left(\frac{\Delta G}{G}\right)^2}, \tag{13}$$

where $w$ (=1 MHz) is the noise equivalent bandwidth of spectrometer channel and $t$ is the observation time. The term $1/wt$ describes a spectrally uncorrelated noise while $(\Delta G/G)$ describes a fully spectrally correlated noise (Jarnot et al., 2006; Ochiai et al., 2013), that is, at first approximation, a radiance offset that is mitigated by the subtraction of the cold sky in





Eq. 11. Considering the average and interpolation on the cold-sky and hot loads outputs, the covariance matrix describing the measurement noise is then:

$$\boldsymbol{S}_y(u,u') = \begin{cases} \sigma_a^2 + \sigma_c'^2/2 + \left(\sigma_c^2 + \sigma_h^2\right)\beta_h^2(u) & \text{if } i = i' \text{ and } j = j' \\ \sigma_c'^2/2 + \left(\sigma_c^2 + \sigma_h^2\right)\beta_h(u)\beta_h(u') & \text{if } |\,i-i'\,| = 1 \text{ and } j = j' \\ \left(\sigma_c^2 + \sigma_h^2\right)\beta_h(u)\beta_h(u') & \text{if } |\,i-i'\,| > 1 \text{ and } j = j' \\ 0 & \text{if } j \neq j' \end{cases} \tag{14}$$

where $u = i \cdot N_f + j$ and $u' = i' \cdot N_f + j'$ and $N_f$ the number of frequencies per spectrum. Here we consider an integration time of 2 sec for the hot load and cold sky spectra in Eq. 10 (used for assessing $\sigma_c$ and $\sigma_h$). The time needed for acquiring the hot-load spectra is available between the termination of a scan and the beginning of the next one. Cold-sky spectra can be obtained in very various ways. A simple one is to construct them using the first 4 and last 4 cold-sky spectra measured during a scan.

The error $e_{NL}$ due to the radiometer non-linear response, i.e., non-zero $\alpha$ in Eq. 9, is the difference between the true brightness temperature $T_b$ and the calibrated one $\widehat{T}_b$, computed as follows (Baron et al., 2011):

1. $G_{\text{cold},i,j}$ is computed applying Eq. (8) to the cold-sky view assuming $C_c = 1800$ ADU that is consistent with Odin/SMR (Olberg et al., 2003), $T_{sys} = 1100$ K, $T_b(\text{cold-sky}) = 0$ K and $\eta_x = 1$. The value $g_{i,j}$ is then computed (Eq. 9).

2. $C_{\text{hot}}$ and $C_{\text{atm}}$ are computed given $T_{hot} = 250$ K and $T_b$ using an iterative process initialized with $G_{\text{cold}}$ (Eqs. 8 and 9).

3. Finally we compute $\widehat{G}$ (Eq. 10), $\widehat{T}_b$ (Eq. 11) and $e_{NL} = \widehat{T}_b - T_b$.

# 4   Retrieval errors

## 4.1   Reference atmosphere

The measurement performances depend on the atmospheric state which depends on the latitude, season and local time. (For our calculations, we assume that the zonal variations of the mean atmospheric state are negligible.) The most relevant parameters to take into account are the profiles of $O_3$, $H_2O$, HCl, temperature and pressure (or geopotential height). A zonal-mean climatology of these parameters has been built, covering all latitudes divided into 11 bins (Fig. 5). These climatologies are based on AURA/MLS observations (v3.3) performed between November 15 2009 and February 15 2010. This period has been chosen because of the strong contrast between the winter-pole and summer-pole conditions that provides large meridional variations of atmospheric states. Moreover, it was characterized by a stable northern polar vortex, which was not affected by any strong perturbation (Kuttippurath and Nikulin, 2012). MLS observes in the moving direction from a sun-synchronous platform. The orbit inclination is 98° from the Equatorial plane. Each latitude is observed at two different local-times, e.g. 1:45 and 13:45 LT at the Equator. These two LT are used to characterize the day- and night-time conditions though it is daytime (nighttime) for both LT over the southern (northern) boreal latitudes (Fig. 2). Bad data have been removed following the MLS user's guide documentation (Livesey et al., 2011), except for the data flagged with negative errors that are biased toward the





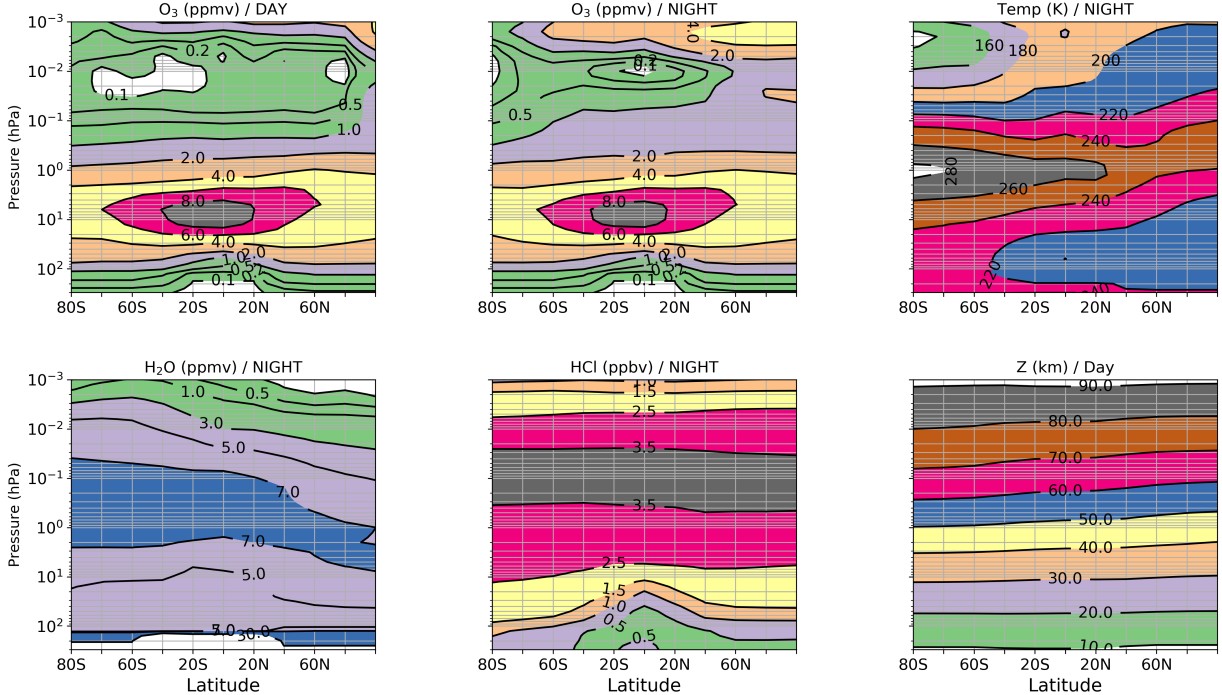

**Figure 5.** Zonal mean distribution of the most relevant atmospheric parameters for the retrieval error assessment. The upper panels show the $O_3$ distributions on pressure levels for day and night labeled climatologies as well as the night temperature one. The lower panels show the night distributions for $H_2O$, HCl, and geopotential altitude. The white regions indicate values smaller than the color scales.

MLS retrieval a-priori. Using such data allows us to span the altitude coverage of the profiles up to 110 km with information suitable for this study. Other molecules are taken from the Whole Atmosphere Community Climate Model (WACCM) (Marsh et al., 2013) and extracted at the climatology latitudes and LT. For HOCl, HCN and CH3CN only tropical profiles are used. Because of their relatively weak signal, their variabilities do not impact the overall measurement performances and only typical
5    abundances are needed to discuss the relevance of the measurement.

### 4.2 Retrieval Method

The simulations are performed with the radiative transfer and retrieval codes used in the SMILES research processing chain (Baron et al., 2011) which has been validated with real observations (Kasai et al., 2013). The retrieved state $\hat{x}$ is a vector including all the unknown parameters of the forward model, namely the atmospheric vertical profiles, a radiance offset on each spectrum
10   and a mean pointing angle offset of the whole scan. The atmospheric profiles are the volume mixing ratio (VMR) profiles of the chemical species, as well as those of temperature and LOS wind.

The retrieval altitudes range from 10 to 90 km, a range fully encompassed within the scan range (10–90 km). The retrieval vertical resolution is 5 km that corresponds to the effective field-of-view of the instrument. Such a setting allows us to perform





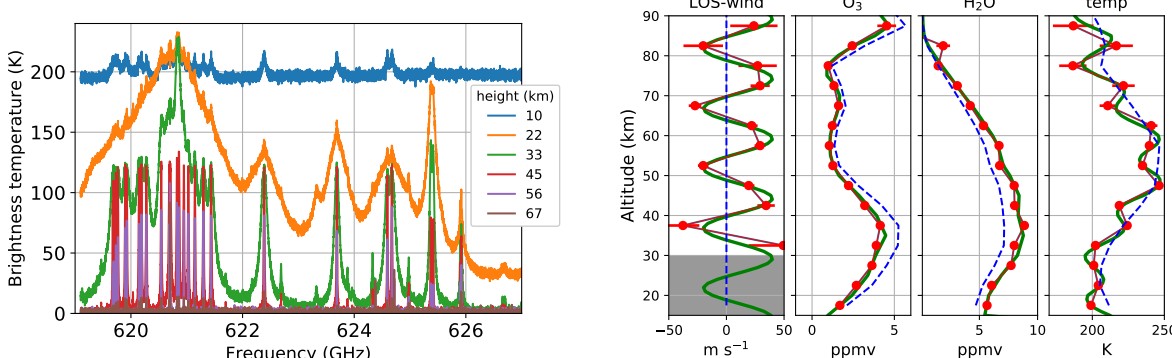

**Figure 6.** Left panel: Double sideband spectra with respect to the lower sideband frequency. Only a few spectra of the full vertical scan are shown (see legend). Right panel: Retrieved profiles with a vertical resolution of 5 km for nighttime arctic conditions. The blue-dashed lines are the a priori profiles (first guess), the green lines are the truth and the red line-circles are the retrieved values. The horizontal bars indicate the 1-$\sigma$ errors due to instrument thermal noise.

retrievals using a simple linear least-squares method with weak regularization. The retrieved vector is given by the equation:

$$\hat{\boldsymbol{x}} = \boldsymbol{x}_0 + (\mathbf{K}^T \mathbf{S}_{d,y}^{-1} \mathbf{K} + \mathbf{U}_x^{-1})^{-1} \mathbf{K}^T \mathbf{S}_{d,y}^{-1} (\boldsymbol{y} - \boldsymbol{y}_0), \tag{15}$$

where $\boldsymbol{y}$ is the measurement, $\boldsymbol{x}_0$ is a first guess of the unknown parameters and $\boldsymbol{y}_0$ is the associated simulated spectra, $\mathbf{K} = \frac{\partial \boldsymbol{Tb}}{\partial \boldsymbol{x}}$ is the Jacobian matrix of the forward model (Eq. 1), $\mathbf{S}_{d,y}$ is a diagonal matrix equal to the diagonal of $\mathbf{S}_y$ (Eq. 14), and $\mathbf{U}_x$ is a

diagonal matrix for stabilizing the matrix inversion. Its element square-roots correspond to very large standard deviations of $\boldsymbol{x}$, typically $> 10000\,\%$, 1000 K, $1000\,\mathrm{m\,s^{-1}}$ for VMR, temperature and LOS wind, respectively. The regularization effects are negligible where the measurement is relevant. The retrieval precision is derived from the linear mapping of the measurement noise covariance onto the retrieved parameters space:

$$\epsilon_{x,\mathrm{n}}^2 = \mathrm{diag}(\mathbf{G}\,\mathbf{S_y}\,\mathbf{G^T}), \tag{16}$$

where $\epsilon_{x,\mathrm{n}}$ is the standard deviation of $\hat{\boldsymbol{x}}$ and $\mathbf{G} = (\mathbf{K}^T \mathbf{S}_{d,y}^{-1} \mathbf{K} + \mathbf{U}_x^{-1})^{-1} \mathbf{K}^T \mathbf{S}_{d,y}^{-1}$.

Figure 6 (right panel) shows retrieved profiles of LOS-wind, $O_3$, $H_2O$ and temperature using a simulated noisy measurement (Fig. 6, left panel). The measurement is computed using perturbed profiles from the climatology at 80°N/nighttime, hereafter named true profiles. The true profiles are defined with a vertical resolution of 0.5 km. The $H_2O$ and HCl climatological profiles are multiplied by 1.2 and the $O_3$ one is multiplied by 0.8. An offset of $-5$ K ($10\,\mathrm{m\,s^{-1}}$) and a 9 km-period oscillation with

an amplitude of 8 K (15 km, $30\,\mathrm{m\,s^{-1}}$) are added on the temperature (wind) profile. A good agreement is found between the retrieved and true profiles. Below 40 km, the wind retrieval error strongly increases and we should consider that 30 km is the lower altitude for wind retrieval. Other profiles are retrieved with low errors over most of the vertical range. A small oscillation is however seen on the $H_2O$ profile that likely arises from the simple retrieval calculation (linearity and weak regularization).





**Table 2.** Systematic errors on observational and forward model parameters: sideband ratio ($w_{\mathrm{lsb}}$ in Eq. 1), calibration hot-load temperature ($T_h$, Eq. 12) and radiance linearity assumption ($\alpha$, Eq. 11), antenna efficiency ($\eta_a$, Eq. 8), spectroscopic line frequency (F), pressure broadening (G) and strength (S) and LOS azimuth and elevation angles ($\theta, \phi$, Fig.4).

| Calibration | | Antenna efficiency | Spectroscopy | | | Sideband ratio | LOS angles |
| --- | --- | --- | --- | --- | --- | --- | --- |
| $\epsilon T_H$ | $\alpha$ | $\eta_a$ | F | G | S | $w_{\mathrm{lsb}}$ | $\theta, \phi$ |
| 1 % | $0.5 \times 10^{-5}$ | $\sim \epsilon T_H$ | 10 kHz | 1 % | 1 % | 1 % | 0.5 mrad |

These results are obtained with relatively large differences between the true and reference profiles and show that this retrieval setting can safely be used for the error analysis.

Systematic retrieval errors emerge from uncertainties on the instrument, calibration and forward model parameters, and LOS angles (Tab. 2). The most critical parameters are investigated using a perturbation method:

$$5 \quad \boldsymbol{\epsilon}_{x,\mathrm{p}} \quad = \quad \mathbf{G}\left(\boldsymbol{y}_{\mathrm{p}} - \boldsymbol{y_0}\right), \tag{17}$$

where $\boldsymbol{\epsilon}_{x,\mathrm{p}}$ is the error induced by the parameter p and $\boldsymbol{y}_{\mathrm{p}}$ is the measurement assessed after changing the value of p according to its uncertainty. It is difficult at this stage of the mission definition to provide proper uncertainties. The values given in Table 2 are relatively close to those expected but rounded in the way that it will be straight-forward to linearly scale the retrieval errors according to any future better knowledge of their uncertainties. One may notice that the uncertainty on the line broadening parameter ($G_i$) is likely underestimated and the actual values should be between 1–4 % depending on the line. On the other hand, the calibration parameters and sideband ratio are likely overestimated. Anyway these errors induce a relatively constant retrieval bias that could be mitigated with ad-hoc corrections if their properties are well understood, e.g., time scale and latitudinal variabilities.

The spectroscopic errors are expressed for each molecule considering that the errors on the line parameters are mutually independent:

$$\boldsymbol{\epsilon}_{x,\mathrm{M}} \quad = \quad \sqrt{\sum_i \left( \boldsymbol{\epsilon}^2_{x,\mathrm{M},F_i} + \boldsymbol{\epsilon}^2_{x,\mathrm{M},G_i} + \boldsymbol{\epsilon}^2_{x,\mathrm{M},S_i} \right)}, \tag{18}$$

where $\boldsymbol{\epsilon}_{x,\mathrm{M}}$ denotes the total spectroscopic error due the molecule M, and $F_i$, $G_i$ and $S_i$ denote the frequency, air-broadening parameter and line-strength of the line $i$.

Several errors will not be discussed in Sect. 5. The errors on the LOS azimuth and elevation angles induces error smaller than $1\,\mathrm{m\,s^{-1}}$ on the LOS wind retrievals. The mean elevation offset of the scan is retrieved with a precision better than $0.2\,\mathrm{mrad}$. The retrieval error induced by the antenna efficiency is not discussed given that it has similar properties than that induced by the hot load emission error (Eq. 12).





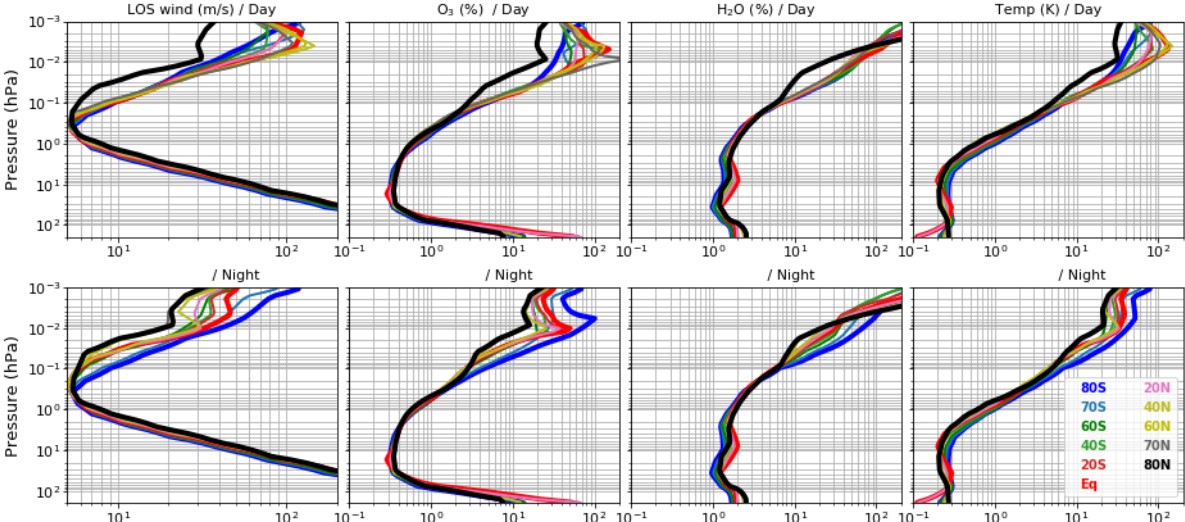

**Figure 7.** Single-scan retrieval precision (1-$\sigma$) for line-of-sight wind (left-most panels), $O_3$ (2nd column panels), $H_2O$ (3rd column panels), and temperature (right-most panels). The line colors correspond to latitude bins (see legend) and thick lines are used for those corresponding to polar and equatorial regions. Errors are given for day- and nighttime labeled profiles. Note that southern (northern) polar profiles are actually both daytime (nighttime) ones.

## 5 Measurement performances

### 5.1 Retrieval precision

Results are discussed on pressure levels and the corresponding altitudes are shown in Figure 5. The precision (1-$\sigma$) is given for a retrieval vertical resolution of 5 km and for a single-scan. For all products except for the LOS-wind, there are 2 quasi-
5  simultaneous and quasi-coincident retrievals available from the two LOS (Fig. 2). They can be averaged for improving the precision by a factor $\sqrt{2}$. Also the retrieval vertical resolution can be decreased for improving the precision of species with weak lines.

### 5.1.1 $O_3$ retrieval

Figure 7 shows the retrieval precisions for temperature, LOS-wind and, $O_3$ and $H_2O$ that have the strongest lines. A good
10  precision is found for $O_3$ retrieval over the whole altitude range (200–0.001 hPa) because of the unusual large number of lines compared to other MM/SMM missions. Between 100 and 0.2 hPa, the relative error is better than 2 % and does not vary significantly with latitudes and local times. A high precision <0.4 % is found between 50–2 hPa. There, the retrieval vertical resolution could be improved to 3–4 km with a precision of $\approx 1$ % (not shown). In the upper part of the retrieval range, the relative precision strongly varies with latitudes and local-times. The errors are 30–50 % in nighttime and 40–100 % in daytime.





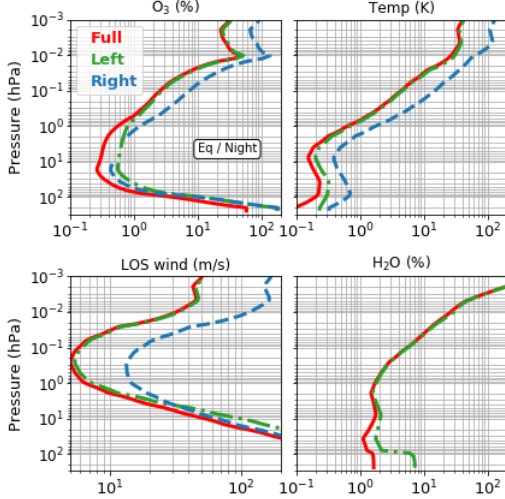

**Figure 8.** Single-scan retrieval precision (1-$\sigma$) for $O_3$ and temperature (upper panels), and line-of-sight wind and $H_2O$ (lower panels). Errors are calculated for a full band retrieval (red lines), the left half IF-band (10.075–14.075 GHz, blue lines) and the right half IF band (14.075–18.075 GHz, green lines). Results are shown for Equatorial nighttime conditions.

The poorest relative precision is found near 0.01 hPa during daytime where most of $O_3$ is photo-dissociated (Fig. 5). Above, the relative precision slightly improves due to the $O_3$ mesospheric secondary peak (Fig. 5).

Figure 8 shows that above 1 hPa most of $O_3$ information is provided from the first half of the spectrum that contains the cluster of $O_3$ lines near 655 GHz (Fig. 3 and Fig. 6). Below this altitude, both sides of the spectrum contribute equally to the $O_3$ retrieval. The $O_3$-line cluster is the main source of information for the LOS wind and temperature retrievals above 4 hPa and 200 hPa, respectively (Fig. 8).

### 5.1.2 Wind and temperature retrievals

The performance of the LOS wind retrieval strongly depends on the $O_3$ abundance. With the current definition of the orbit (equator ascending node at 18:00 LT), most of the measurements are performed in nighttime (Fig. 2), which is a favorable case for measuring wind. The best performances are found over the northern polar region where LOS-wind can be retrieved with a precision better than 10 m s$^{-1}$ between 2 and 0.02 hPa (Fig. 7). Comparable performances are found for nighttime equatorial and mid-latitude retrievals over a similar vertical range but with a slightly lower upper limit (0.03–0.04 hPa). In daytime, the uppermost altitude for obtaining similar precision dropped to 0.1 hPa over most of the latitudes. At 10 hPa, the error is 50–60 m s$^{-1}$ and averaging 2 weeks of equatorial data in 10° latitude bin gives a precision of about 2 m s$^{-1}$. Since the precision is much poorer below this altitude, the 10 hPa level should be considered as the lowest altitude for obtaining useful wind information.





At 0.01 hPa, the nighttime LOS wind precision changes with latitudes from $20\,\mathrm{m\,s^{-1}}$ to $50\,\mathrm{m\,s^{-1}}$ (the southern polar profile is excluded) and from $40$–$60\,\mathrm{m\,s^{-1}}$ in daytime. At this altitude, the $H_2O$ line at 620.7 GHz contributes significantly to the wind retrieval, especially during daytime. Over the polar regions, strong NO enhancements frequently occur in the middle atmosphere due to energetic particle precipitation (EPP) (Randall et al., 2007; Pérot et al., 2014; Orsolini et al., 2017). During

such events, the NO lines can be increased by more than a factor 10 that would improve the wind and temperature retrievals.

Temperature can be retrieved with a precision better than 1 K below 1 hPa. The retrieval vertical resolution can be improved to 3 km with a precision better than 1 K between 200 and 5 hPa (not shown). Above 0.2 hPa, the precision decreases to 10–30 K near 0.01 hPa in nighttime and to 30–80 K in daytime. During daytime most of the mesospheric information is provided by the strong $H_2O$ line at 620.7 GHz.

### 5.1.3  $H_2O$ and other molecules retrievals

The $H_2O$ profile is retrieved from the line at 620.7 GHz and, below 100 hPa, from the continuum induced by far lines. The precision is better than 3 % (20 %) below 0.3 hPa (0.05 hPa). For altitudes above 0.1 hPa, the relative error increases and exhibits large latitudinal variations, e.g., 10–50 % at 0.2 hPa. The largest errors are found during daytime when the signal from $O_3$ is weak. Under such conditions, temperature is retrieved from the single $H_2O$ line. The forward model inversion

becomes ill-conditioned and both $H_2O$ and temperature errors strongly increase. This issue is clearly illustrated with the much smaller $H_2O$ daytime errors estimated without retrieving temperature compared to those with temperature retrieval (Fig. 9). Constraining the mesospheric temperature would significantly improve the mesospheric $H_2O$ retrievals.

The retrieval precision for other molecules are shown in Fig. 9. First let's note that except for $O_3$ and $H_2O$, all chemical species are retrieved from optically thin lines and the VMR error profiles have similar characteristics and are independent of

the VMR values. The minimum VMR error is found near 10 hPa. Below, the errors increase due spectral line overlaping. The atmosphere becomes opaque near the tropopause. From the middle stratosphere to the mesosphere, the errors increase due to the decrease of atmospheric density (the error decreases only as $\approx$ pressure$^{-0.6}$ because the density decrease is partly compensated by the narrowing of the lines).

The best measurement performances with respect to the VMR are found for HCl, $N_2O$, HCN, CH3CN and $HNO_3$. Good

information can also be inferred for the four most abundant $O_3$ isotopologues and from HDO. Important chemically active species such as ClO, NO, $NO_2$ or $HO_2$ can also be retrieved. If necessary the relative precision can be improved by averaging profiles or decreasing the retrieval vertical resolution. Deriving useful information for species such as BrO or HOCl will be challenging.

Chemically active species exhibit large variabilities. The photo-chemistry driven diurnal variation is the most common one.

For instance, stratospheric ClO, NO and mesospheric $HO_2$ are more abundant in daytime but vanish during nighttime. Special events that occur more or less frequently can strongly increase the signal-to-noise ratio. For instance, ClO VMR frequently reaches 1.5 ppbv near 20 km during polar springtime due to the chlorine activation during the polar winter. The enhancement of $SO_2$ after strong volcanic eruption can also be measured (Pumphrey et al., 2015). EPP induced enhancement of $NO_x$ and $HO_x$ are an other example. During such events nighttime NO can reach levels of 10-100 ppbv between 10–0.1 hPa, levels





much larger than the measurement single-scan precision (2–20 ppbv). EPP induced enhancements are not well represented in the models (Randall et al., 2007; Pérot et al., 2014; Orsolini et al., 2017) and SIW has a strong potential for providing key insights on their dynamical and chemical sources.

## 5.2 Systematic errors

The errors induced by the spectroscopic uncertainties on the most important lines have been estimated for the LOS-wind, temperature, $O_3$, $H_2O$ and HCl retrievals. We consider the 50 most intense $O_3$ lines over the whole bandwidth, two HCl triplets (624.9 and 625.9 GHz), two NO triplets (651.4 and 651.7 GHz) and the 620.7 GHz $H_2O$ line. Systematic errors induced by the double-sideband ratio (DSB), the calibration hot-load temperature and the radiometer non-linearity are also discussed for the same products.

### 5.2.1 Wind retrieval

Figure 10 clearly shows three altitude ranges for the wind retrieval errors induced by the spectroscopic parameters. Results are given for the latitude 60°N. Above 0.1 hPa, a daytime error of 3–5 $\mathrm{m\,s^{-1}}$ is induced by the frequency uncertainty on the $H_2O$ line (Tab. A3). During nighttime, the signal is dominated by about 15 $O_3$-lines and a retrieval error of only 1 $\mathrm{m\,s^{-1}}$ is induced by their frequency uncertainty (Tab. A1 and A2). The same error of 1 $\mathrm{m\,s^{-1}}$ is found between 1 and 0.1 hPa both during day
and night times. No impact of the NO lines have been found, even at higher latitudes, but this would not be the case for EPP enhanced profiles. Below 1 hPa, the lines broadened by the pressure overlap over each other. Consequently the uncertainties on the air-broadening parameters and to a lesser extent, the line strength of the $O_3$ lines contribute to the retrieval error. The bias increases up to 20–30 $\mathrm{m\,s^{-1}}$ at 10 hPa.

Figure 11 shows the retrieval errors induced by the double-sideband ratio (DSB), the calibration hot-load temperature and
the radiometer non-linearity. Those parameters introduce errors on the wind retrieval only below 2 hPa. Above 10 hPa, errors are dominated by the DSB uncertainty that can be larger than 20 $\mathrm{m\,s^{-1}}$. Other errors are lower than 10 $\mathrm{m\,s^{-1}}$.

The $O_3$ lines parameters and the DSB are the dominant sources of uncertainties between 10–1 hPa for all latitudes (Fig. 12). A preliminary study shows that the DSB can be characterized with an error better than 0.1 % over the entire bandwidth. If this is confirmed the DSB induced error would be significantly less than that estimated here. On the other-hand errors induced
by the line air-broadening parameters may be underestimated. Different methods can be used to reduce the wind retrieval biases between 5–10 hPa. For instance, an observed 20–40 $\mathrm{m\,s^{-1}}$ bias between 8–5 hPa on JEM/SMILES wind retrievals was reduced to less than 4 $\mathrm{m\,s^{-1}}$ between 30S–50N by considering that the mean tropical flow was zonal (Baron et al., 2013b). Meteorological analyses and reanalysis at mid-latitudes can also be used for characterizing the retrieval biases below 5 hPa.

### 5.2.2 Temperature and VMR retrievals

The biases on $O_3$, HCl and temperature retrievals due to the spectroscopic parameters are small. They are lower than 1 % and 0.5 K between 100 and 0.02 hPa. Above 0.02 hPa, the biases increase but remain smaller than 5 % and 4 K. The errors are







**Figure 9.** Retrieval single-scan precision (full lines), nighttime VMR (dashed lines) and daytime VMR (dotted lines) profiles. Profiles are shown at 80°S (blue line), Equator (red line) and 80°N (black line). The thick (thin) red full-lines are nighttime (daytime) conditions. The H$_2$O results without temperature retrieval are indicated with "w/o T".





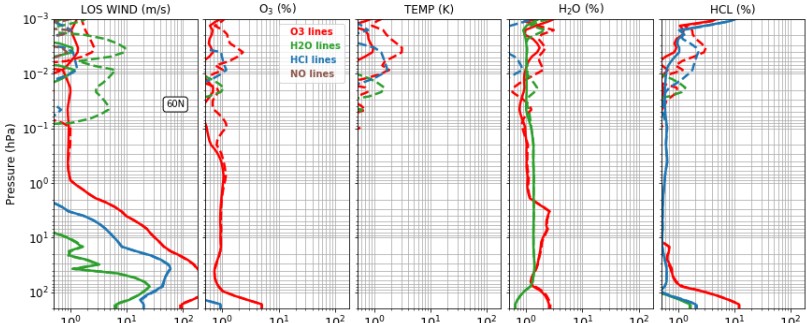

**Figure 10.** Spectroscopic induced errors on LOS wind, temperature, $O_3$, $H_2O$ and HCl retrievals (see panel titles). The full-lines (dashed-lines) show the nigthtime (daytime) condition at 60N.

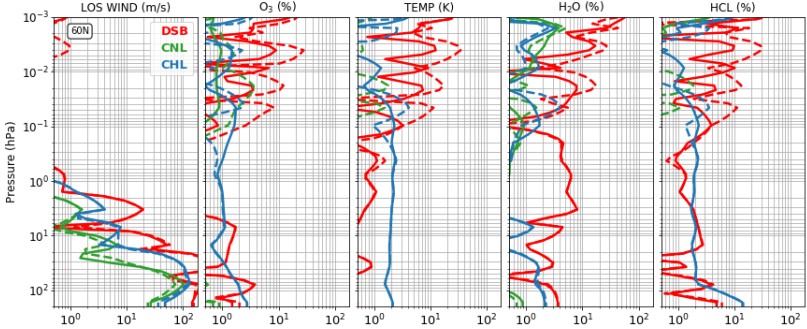

**Figure 11.** Same as Fig. 10 but for double-sideband ratio (DSB), radiometer nonlinearity (CNL) and calibration hot-load emission (CHL).The error assumptions are summarized in Tab. 2.

induced by the air-broadening and strength parameters of $O_3$ and $H_2O$ lines. The strong impact of the $H_2O$ line parameters onto the HCl retrieval reveals error amplifications due to the temperature retrieval. Using constraints on the temperature retrieval should allow us to reduce such effects.

The retrieval of $H_2O$ above 0.2 hPa has a small bias <2 % that is induced by the uncertainties on the air-broadening and

5 strength parameters of the 620.7 GHz $H_2O$ line. Below this altitude, the retrieval error reaches 5 % mainly due to the air-broadening parameters of the overlapping $O_3$ lines at 620.69 and 623.669 GHz (Tab. A1). Below 100 hPa, the $H_2O$ lines outside the band are the main signal for the retrieval (not shown), and the 620.7 GHz $H_2O$ line weight in the retrieval is small.

The DSB and calibration parameters induced errors on $O_3$, HCl and temperature retrieval are small below 0.1 hPa, i.e., lower than 3 % and 2 K (Fig. 11). The calibration hot-load temperature and the radiometer non-linearity dominate the tem-

10 perature retrieval error. The VMR retrievals are also sensitive to the DSB uncertainties, especially for the $H_2O$ retrieval in the stratosphere and lower mesosphere. In the mesosphere, the DSB uncertainty generates significant retrieval errors on the VMR and temperature profiles, uncertainties that are larger in daytime (increase likely due to the temperature retrieval). However as we previously discussed for the wind retrieval, the DSB uncertainty is likely overestimated by a factor 10.





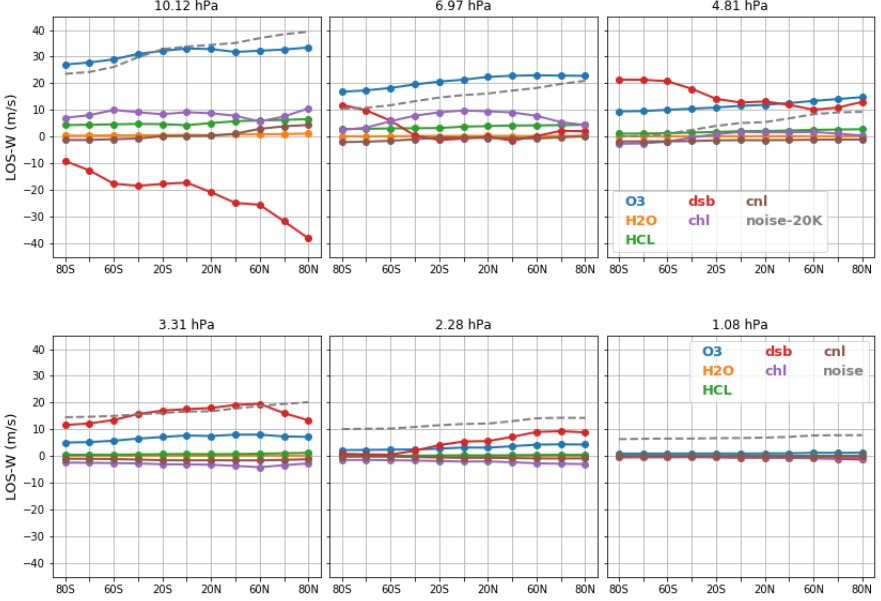

**Figure 12.** Line-of-sight wind retrieval biases with respect to latitudes near 10, 7 and 5 hPa (upper panels), and near 3, 2 and 1 hPa (lower panels). Biases are shown for the uncertainties on the spectroscopic parameters of $O_3$, HCl, and $H_2O$, the double-sideband ratio, the calibration hot load and the calibration non-linearity.

## 6 Conclusions

A simulation study has been conducted to support the mission definition of SIW and to assess the measurement performances. This small mission will be launched near 2022 for monitoring the middle-atmosphere (10–90 km) using the thermal emission lines near 640 GHz of a large number of chemical species. This analysis focuses on the main outcomes, namely LOS wind,
5   temperature, $O_3$ and more than a dozen of other chemical species. The error assessment is performed taking into account the day-night and latitudinal atmospheric variabilities.

    The unusual large number of strong $O_3$ lines at 653–657 GHz allows us to measure the 2-d horizontal wind between 10 and 0.001 hPa and temperature between 100 and 0.1 hPa as well as providing a high-sensitivity to $O_3$ between 100–0.001 hPa. LOS wind is an original outcome for such a mission. It demands a special observation setting involving 2 antennas in order
10  to retrieve two perpendicular components of the wind vector. Each component can be measured between 2 and 0.03 hPa with a precision better than $10\,\mathrm{m\,s^{-1}}$ and vertical resolution of 5 km. Other space-borne instruments have poor sensitivity in this altitude range. A sunsynchronous polar orbit allowing us to perform night-time measurements is currently considered. Such conditions are the most favorable for mesospheric wind, temperature and ozone measurements but not for active chemical species such as stratospheric ClO or strato-mesospheric $HO_2$ that vanish during nighttime.





The impact of systematic errors induced by the spectroscopic parameters and, by the instrument and calibration parameters are discussed. This work highlights the need for a good characterization of the radiometer sideband ratio and of the spectroscopic parameters (air-broadening and strength) of key $O_3$, $H_2O$ and $NO$ lines. Even so, a large wind measurement bias may occur between 10 and 2 hPa mainly due to errors on $O_3$ line air-broadening parameters. The need for developing ad-hoc
methods for reducing this bias must also be studied.

SIW shows a strong potential for various scientific issues. It can provide for the first time global information on the horizontal wind between 30–90 km that can be used to validate chemical and climate models. It has the potential for contributing to the characterization of long trend series of temperature, $O_3$, $H_2O$ and $HCl$, that are important for climate studies and for monitoring the chemical composition of the mid-atmosphere. The mission can provide data to study the dynamics of the middle
atmosphere. Based on SIW observations, specific studies on key dynamical processes, such as the quasi-biennial oscillation, the semi-annual oscillation or sudden stratospheric events for example, could be carried out. A better understanding of these phenomena, in addition to global mid-atmospheric wind measurements, would significantly improve our knowledge of the climate system. Not discussed in this paper is also the capability of SIW for measuring ice water content in the tropical upper-troposphere (Eriksson et al., 2014). Observing the same air-mass from two perpendicular directions could provide interesting
information considering the high spatial inhomogeneities of cloudy scenes.

Optimization of the calibration procedure will be studied in order to improve the measurement precision. Here we have assumed an equal observation time for the cold-sky and atmosphere measurements. Changing the time sharing in favor of atmospheric observations could improve the measurement sensitivity by more than 20 %. The InnoSat platform offers a quick opportunity to fly SIW. This is important since the current SMM limb missions have already exceeded by far their lifetime
expectations and they risk to stop soon. However such a platform strongly limits the design of a SMM instrument and its performances. A larger antenna would improve the vertical resolution and an additional receiver with a narrow bandwidth measuring an oxygen or a strong water vapor line would significantly improve the wind and temperature retrievals in the mesosphere. Such improvements are studied for the much larger SMILES-2 mission (Ochiai et al., 2017) presented in the introduction section and which also includes the same spectral window as SIW. However this mission can not be launched
before 2024/25.

## Appendix A: LOS and horizontal winds

Wind measure two components of the horizontal wind vector over the same region, allowing us to compute the meridional (V) and zonal (U) components. Applying Eq. (6) to the forward and aft viewing antenna, the two retrieved LOS wind are:

$$V_{\mathrm{los,fwd}} = V\cos(\phi_n) + U\sin(\phi_n)$$
$$V_{\mathrm{los,aft}} = V\cos(\phi_n+\delta) + U\sin(\phi_n+\delta) \tag{A1}$$





where $\alpha_n$ is the angle of the forward-looking line-of-sight with respect to the north direction and $\delta$ is the angle between the two lines of sight. It is straight forward to show that

$$U = \frac{1}{\sin(\delta)}\left(V_{\text{los,aft}}\cos(\phi_n) - V_{\text{los,fwd}}\cos(\phi_n + \delta)\right) \qquad \text{(A2)}$$

$$V = \frac{1}{\sin(\delta)}\left(V_{\text{los,fwd}}\sin(\phi_n + \delta) - V_{\text{los,aft}}\sin(\phi_n)\right)$$

and the errors on $U$ and $V$ are:

$$\epsilon_U = \frac{\epsilon_{\text{los}}}{\sin(\delta)}\sqrt{\cos(\phi_n+\delta)^2 + \cos(\phi_n)^2}$$

$$\epsilon_V = \frac{\epsilon_{\text{los}}}{\sin(\delta)}\sqrt{\sin(\phi_n+\delta)^2 + \sin(\phi_n)^2} \qquad \text{(A3)}$$

where $\epsilon_{\text{los}}$ is line-of-sight wind retrieval error. For the $\delta = 90°$, i.e. this analysis assumption, we have

$$U = \left(V_{\text{los,aft}}\cos(\phi_n) + V_{\text{los,fwd}}\sin(\phi_n)\right)$$

$$V = \left(V_{\text{los,fwd}}\cos(\phi_n) - V_{\text{los,aft}}\sin(\phi_n)\right)$$

and the error on each wind component becomes $\epsilon_U = \epsilon_V = \epsilon_{\text{los}}$. The transformation $(V_{\text{los,fwd}}, V_{\text{los,aft}})$ to $(U, V)$ corresponds to a vector rotation of $\frac{\pi}{2} - \phi_n$. This configuration is that for which $\epsilon_U^2 + \epsilon_V^2 = 2\left(\epsilon_{\text{los}}/\sin(\delta)\right)^2$ is minimum.

**Appendix B: Spectroscopic lines**

The following tables show the most relevant spectroscopic lines for the retrievals of the LOS wind, $O_3$, Temperature, $H_2O$ and HCl. The relative retrieval impact of each parameter is defined as:

$$\varrho_{x,\text{M},p_i} = \frac{\epsilon_{x,\text{M},p_i}}{\max(\epsilon_{x,\text{M}})} \quad \text{with } p_i = F_i, G_i \text{ or } S_i, \qquad \text{(B1)}$$

and $\epsilon_{x,\text{M},p_i}$ and $\epsilon_{x,\text{M},p}$ are the errors from a single parameter and all molecule lines (Eq. 18).

*Competing interests.* The authors declare that they have no conflict of interest

*Acknowledgements.* Omnisys Instruments (Sweden) has designed the instrument and provided valuable information and discussion for this study. WACCM data has been provided bu Yvan Orsolini (NILU, Norway), Varavut Limpasuvan (Coastal Carolina University, USA) and Naohiro Manago (Chiba University, Japan) .





**Table A1.** Relative impact of $O_3$ line parameters on temperature, $O_3$, $H_2O$ and LOS wind retrievals. For a retrieved product, the impact is defined as error to the maximum error ratio (see text). Results are given for 10, 1 and 0.1 $\mathrm{hPa}$ levels (Equatorial night-time conditions). Only parameters having an impact larger than 0.5 at any of the considered altitudes are shown. The parameters are the line frequency (S), the air-broadening parameter (G) and the line strength (S). The line is characterized by its frequency (MHz).

| Parameter | LOS wind | | | O3 | | | Temperature | | | H2O | | | HCl | | |
|---|---|---|---|---|---|---|---|---|---|---|---|---|---|---|---|
| | | | | | | Lower sideband | | | | | | | | | |
| 620687-F | - | - | 0.6 | - | - | - | - | - | - | - | - | - | - | - | - |
| -G | - | -0.5 | - | - | - | - | - | - | - | 0.8 | 0.5 | - | - | - | - |
| -S | - | **-1.0** | - | - | - | - | - | - | -0.5 | **1.0** | 0.7 | - | - | - | - |
| 620825-F | - | 0.6 | **1.0** | - | - | - | - | - | - | - | - | - | - | - | - |
| -G | 0.8 | - | - | - | - | - | - | 0.9 | - | 0.7 | - | - | - | - | - |
| -S | 0.7 | - | - | - | - | - | - | **1.0** | 0.8 | 0.6 | - | - | - | - | - |
| 623688-F | - | 0.8 | 0.7 | - | - | - | - | - | - | - | - | - | - | - | - |
| -G | - | - | - | - | - | - | 0.8 | - | - | -0.7 | -0.5 | - | - | - | - |
| -S | - | - | - | - | - | - | 1.0 | - | - | -0.8 | -0.5 | - | - | - | - |
| 625370-F | - | 0.7 | - | - | - | - | - | - | - | - | - | - | - | - | - |
| -G | -1.0 | - | - | - | - | - | - | - | - | - | - | - | - | - | - |
| -S | - | - | - | **1.0** | **1.0** | **1.0** | **-1.0** | - | 0.5 | - | **1.0** | **1.0** | - | **1.0** | **1.0** |
| | | | | | | Upper sideband | | | | | | | | | |
| 650732-F | - | 0.7 | - | - | - | - | - | - | - | - | - | - | - | - | - |
| -G | **-1.0** | - | - | - | - | - | - | - | - | - | - | - | - | - | - |
| -S | -0.5 | - | - | 0.9 | 0.9 | 0.9 | -0.9 | - | - | - | 0.9 | 0.9 | - | 1.0 | 0.9 |
| 651475-F | - | 0.7 | 1.0 | - | - | - | - | - | - | - | - | - | - | - | - |
| -G | -0.5 | - | - | - | - | - | - | - | - | - | - | - | 1.0 | - | - |
| -S | -0.6 | - | - | - | - | - | - | - | - | -0.5 | - | - | **1.0** | - | - |
| 651556-F | - | 0.9 | 0.5 | - | - | - | - | - | - | - | - | - | - | - | - |
| -S | - | - | - | - | - | - | - | - | -0.9 | - | - | - | - | - | - |
| 653763-F | - | 0.7 | 0.9 | - | - | - | - | - | - | - | - | - | - | - | - |
| -G | - | - | - | - | - | - | - | - | - | - | - | - | -0.6 | - | - |
| -S | - | - | - | - | - | - | - | - | - | - | - | - | -0.6 | - | - |



**Table A2.** Continuation of Tab. A1.

| Parameter | LOS wind | | | O3 | | | Temperature | | | H2O | | | HCl | | |
|---|---|---|---|---|---|---|---|---|---|---|---|---|---|---|---|
| 654713-F | - | 0.7 | 0.9 | - | - | - | - | - | - | - | - | - | - | - | - |
| -S | -0.5 | - | - | - | - | - | - | - | - | - | - | - | - | - | - |
| 654851-F | - | 0.7 | 0.9 | - | - | - | - | - | - | - | - | - | - | - | - |
| 655004-F | - | 0.7 | 0.8 | - | - | - | - | - | - | - | - | - | - | - | - |
| 655121-F | - | 0.7 | 0.8 | - | - | - | - | - | - | - | - | - | - | - | - |
| 655202-F | - | 0.7 | 0.9 | - | - | - | - | - | - | - | - | - | - | - | - |
| 655289-F | - | 0.6 | 1.0 | - | - | - | - | - | - | - | - | - | - | - | - |
| -G | - | - | - | - | - | - | - | 0.5 | - | - | - | - | - | - | - |
| -S | - | - | - | - | - | - | - | 0.6 | 0.8 | - | - | - | - | - | - |
| 655607-F | - | 0.7 | 0.9 | - | - | - | - | - | - | - | - | - | - | - | - |
| -S | - | - | - | - | - | - | - | - | - | - | 0.5 | - | - | - | - |
| 655873-F | - | 0.7 | 0.9 | - | - | - | - | - | - | - | - | - | - | - | - |
| 655960-F | - | 0.7 | 0.9 | - | - | - | - | - | - | - | - | - | - | - | - |
| 656005-F | - | 0.8 | 0.8 | - | - | - | - | - | - | - | - | - | - | - | - |
| -S | - | - | - | - | - | - | - | - | 0.5 | - | - | - | - | - | - |
| 656224-F | - | 0.7 | 0.8 | - | - | - | - | - | - | - | - | - | - | - | - |
| 656251-F | - | 0.7 | 0.7 | - | - | - | - | - | - | - | - | - | - | - | - |
| 656383-F | - | 0.8 | 0.6 | - | - | - | - | - | - | - | - | - | - | - | - |
| -S | - | - | - | - | - | - | - | - | -0.6 | - | - | - | - | - | - |
| 656419-F | - | 0.8 | 0.5 | - | - | - | - | - | - | - | - | - | - | - | - |
| -S | - | - | - | - | - | - | - | - | -0.9 | - | - | - | - | - | - |
| 656461-F | - | 0.7 | - | - | - | - | - | - | - | - | - | - | - | - | - |
| -S | - | - | - | - | - | - | - | - | **-1.0** | - | - | - | - | - | - |
| 656476-F | - | 0.6 | - | - | - | - | - | - | - | - | - | - | - | - | - |
| -S | - | - | - | - | - | - | - | - | -0.7 | - | - | - | - | - | - |

**Table A3.** Same as Tab. A1 but for the $H_2O$ line parameters

| Parameter | LOS wind | | | O3 | | | Temperature | | | H2O | | | HCl | | |
|---|---|---|---|---|---|---|---|---|---|---|---|---|---|---|---|
| 620701-F | - | **1.0** | **1.0** | - | **-1.0** | - | - | - | - | - | - | - | - | 0.5 | - |
| -G | **1.0** | **-1.0** | - | 0.9 | 0.8 | **1.0** | 0.6 | - | **1.0** | 0.9 | 0.9 | 0.7 | 0.7 | - | **1.0** |
| -S | - | - | - | **1.0** | 0.7 | - | **1.0** | **1.0** | - | **1.0** | **1.0** | **1.0** | **1.0** | **1.0** | - |
| 622482-G | - | - | - | - | -0.5 | - | - | - | - | - | - | - | - | - | - |
| -S | -0.8 | - | - | - | 0.7 | - | - | - | - | - | - | - | - | - | - |





**Table A4.** Same as Tab. A1 but for the HCl line parameters.

| Parameter | LOS wind | | | O3 | | | Temperature | | | H2O | | | HCl | | |
|---|---|---|---|---|---|---|---|---|---|---|---|---|---|---|---|
| 624964-S | -0.6 | - | - | - | - | - | -0.6 | 0.5 | - | - | - | - | - | - | - |
| 624977-G | - | - | - | - | - | - | 0.6 | - | - | - | - | - | - | - | - |
| -S | - | - | - | - | - | - | -1.0 | 0.5 | - | - | - | - | - | - | - |
| 625901-F | - | 0.6 | 0.6 | - | - | - | - | - | - | - | - | - | - | - | - |
| -G | 0.9 | - | - | -0.7 | -0.7 | -0.6 | -0.6 | -0.9 | - | - | -0.6 | -0.6 | - | - | - |
| -S | **-1.0** | - | - | - | - | - | **1.0** | - | 0.5 | - | - | - | 0.7 | 0.5 | 0.6 |
| 625918-F | - | **1.0** | **1.0** | - | - | - | - | - | 0.5 | - | - | - | - | - | - |
| -G | - | - | - | **-1.0** | **-1.0** | **-1.0** | - | **-1.0** | **1.0** | **-1.0** | **-1.0** | **-1.0** | - | -0.5 | - |
| -S | 1.0 | - | - | - | - | - | 0.6 | -1.0 | 0.6 | 0.7 | - | - | **1.0** | **1.0** | **1.0** |
| 625931-G | -0.6 | - | - | - | - | - | - | -0.5 | - | - | - | - | - | - | - |
| -S | **1.0** | - | - | - | - | - | - | - | -0.7 | - | - | - | - | - | - |



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
