# Peer review of "Simulation study for the Stratospheric Inferred Winds (SIW) sub-millimeter limb sounder"

_Atmospheric Measurement Techniques, 2018_

## Referee Comment (RC1) · Anonymous Referee #1 · 17 Apr 2018

**Reviewer report: P. Baron et al. 2018**

**General comments**

The paper describes a simulation study for a future InnoSat instrument. It focusses on wind, temperature and trace gas concentration observations in the middle-atmosphere a region where such space-borne observations are missing (wind) or expected to be lacking in near future due to the end of lifetime of other satellites (trace gases). The study is therefore of high scientific relevance and perfectly suited to the scope of AMT. The research lies on solid grounds and is generally well presented in the manuscript. A few modifications pending I suggest publication in AMT. My most important questions are the following:

- Although primarily focusing on wind retrievals the manuscript does not adress the question of local oscillator stability. However, for a wind error below 1 m/s an oscillator stability of $\Delta\nu/\nu < 3.3\cdot10^{-9}$ is needed. As the operation geometry does not allow observations in opposite viewing directions as done by other wind measurement techtiques this stability needs to be long-term (i.e. for the entire mission lifetime) in order not to introduce trend artefacts. What are the expected frequency stability during heating/cooling of the satellite? What is the aging of the planned frequency source? Are there plans for a method to monitor the LO frequency? Please comment on this issue in the manuscript. Even if very high stability could be achieved and and the induced bias was marginal, an upper limit should be indicated. I assume it is one of the first questions a reader has when reading about Doppler shift measurements.

- You suggest a sun-synchronous orbit crossing the equatorial ascending node at 18:00. I agree that this allows perfect conditions for wind observations by allowing to constantly observe the night side with the higher ozone concentrations. My concern about this choice is that the representativeness of the measurements of trace gases near the day-night terminator may be delicate as during this period the concentrations of photochemically active species undergo rapid changes. This may among others introduce an artificial annual cycle to your measurements simply by modifying the time before (after) the sunrise and after (before) the sunset the observations are made. Can you quantify this effect and what are your ideas to mitigate it?

**Specific comments**

p.1, l.1: Why is your instrument called "Stratospheric Inferred Winds" when effectively assessing the upper stratosphere and lower mesosphere?

p.2, l.5: "risk of an observation gap in the near future" Is there no citation for this statement?

p.2, l.20: You could maybe add a citation of a modeler stating that wind simulations in these regions are hard to obtain.

p.2, l.24: does → do

p.2, l.25/27: The measurement approach presented in Baumgarten 2010 is also providing wind in the gap region. Please add citation at l.27.

p.2, l.30: Rather 20 or 30 km? Do you see Aeolus as a good complement to your mission if you get synchronous mission activity? Please comment on this.

p.3, l.1: Please clarify that the wind profiles published in Wu et al. 2008 do not cover the gap region you defined as 30-70 km.

p.3, l.9: Please indicate the expected lifetime of SIW. Is there a chance to have it observing at the same time as SMILES-2? Would there be an added-value if both missions would be observing synchronously or would the expected higher performance of SMILES-2 make SIW obsolete?

Sect. 2.1: Please extend the instrument description. It is clear that this is not an instrument paper, but some core characteristics of the receiver should be introduced here. This will moreover avoid that questions arise during the further read of the manuscript.

p.4, l.6: With your scanning scheme LOS winds at 45 and 134° will be recorded from a similar location only for 1 altitude of your scan. How large will the distance between this two components be at maximum? Is this sampling mismatch not critical for your calculations of one zonal and meridional wind profile with Eq. (A2)?

p.4, l.7: "continuously rotate" is misleading as it is in fact not an unaccelerated rotation but rather a succession of upward and downward scans.

p.4, l.28: "at least a factor of 2... compared to other spectral regions." Please be more concrete. What are "other spectral regions".

Fig. 3: You display 9 GHz but the spectrometer bandwidth is 8 GHz. Please mark the (un)used frequency range in this figure.

Fig. 3: What is the reason for the 2 GHz frequency shift compared to SMILES-2 (Ochiai et al 2017)?

Fig. 3: Displaying the centre frequecy of LSB and USB directly in the different panels would help to further clarify the figure.

p.5, l.7: I suggest to modify "so-called brightness temperature" to "so-called Rayleigh-Jeans brightness temperature" to make sure the user is not confused by the 1 mK of cosmic background (as I was at first).

p.6, l.6: $i \rightarrow \nu_i$

Fig. 4: It is not completely clear from the text what you intend to communicate to the reader with this figure. Please extend the description and reasoning in the text.

Eq. (2): Why do you use quadratic addition of the static antenna pattern and the broadening due to the scanning velocity? I would argue that, if you combine the static pattern with the scanning, your beam pattern becomes non-Gaussian. In any case, I think that by quadratic addition you drastically underestimate the width of your main lobe unless the scanning broadening is much smaller than the static beam width. If not the case, I would think that a linear addition would already be closer to reality while still underestimating the width of your beam (due to the non-Gaussicity introduced by the scanning). Please review the information about the beam width in the manuscript.

p.8, l.14: Why are you using JPL for some lines and Hitran for others?

p.6, l.16/17: You could indicate worst-case bias induced by these effects to show how marginal they are.

p.10, l.1: You refer to $T_{so}$ as antenna spillover. Looking at Eg (8) $T_{so}$ rather refers to the average brightness temperature of the regions where the radiation which you receive because of the spillover in your optics actually comes from. Please adapt the wording.

p.10, l.2/l.15: "spill-over" or "spillover". Please use consistent spelling

Eq. (10): You may state that you assume $T_c = 0$ here.

Eq. (15): Does this linear approach suffice for all situation you expect to encounter? What happens if the truth is further away from the first guess than in Fig. 6? Is this linear retrieval also sensible for photochemically active species close to the day/night terminator?

Eq. (15): You use $S_{d,y}$ instead of $S_y$. What about error correlations? Can they be neglected and why? Pleas state in text.

p.13, l.15: Please add the reason for the increasing error at lower altitudes here.

p.14, l.19: I suggest to refer to $F_i$ as "centre frequency" instead of just "frequency".

p.13, l.20: How do you retrieve the elevation offset?

p.16, l.10: Please indicate the reason why the best performance is found over the northern polar regions.

Sect. 5.2: Why using a 10 times to large error for the sideband ratio? Indeed a 1% sideband uncertainty seems rather large. Please consider to modify it to the value of 0.1% that you found in your preliminary study. The choice of 1% uncertainty which you then qualify several times as too large unnecessarily complicates the reading of this section.

p.18, l.15: have → has

p.18, l.16: overlap over each other→ overlap each other

p.21, l.7: unusual → unusually

p.22, l.28: Gramatically incorrect sentence

p.23, l.1: $\alpha_n \rightarrow \phi_n$

Eq. (B1): What is the significance of "max" here? $\epsilon_{x,M}$ seems to be a scalar (see Eq. (18)) so I don't see what you want to do by taking the maximum.

---

## Referee Comment (RC2) · Anonymous Referee #2 · 16 Jun 2018

Review of: Simulation study for the Stratospheric Inferred Winds (SIW) sub-millimeter limb sounder

Baron et al. ATMD

—- General comments

This is a very nice paper that does a thorough job of describing the theoretical capabilities of an exciting new instrument under development. I have no hesitation in recommending that this paper be published in pretty much its current form, pending the authors addressing the mostly minor comments I have below.

My only broad comment is that I think the authors have somewhat glossed over the trade inherent in such measurements between precision and resolution (vertical res-

[Figure]

olution mainly in this case). They can afford not to dwell on it in this study because they've chosen a rather coarse 5km spacing for their state vector resolution, but I do think it deserves some mention (but probably not any new calculations). At no point do they discuss averaging kernels, but I imagine they've computed them, or could do so very easily. I expect that such kernels indicate that the information content is a good match for their 5 km grid (i.e., that A is a good approximation to the identity matrix, except perhaps at the top and the bottom of the profiles). I would however like them to add a discussion to that effect somewhere in the text. Their mention of the potential of finer 3km resolution for temperature (page 17, line 7) implies that they have looked into this issue somewhat. I encourage them to talk about it just a bit more.

One other minor point. The authors make no mention of frequency stability requirements for the instrument Local Oscillator (presumably tied to some lower frequency clock source). I presume the instrument (or spacecraft) design includes some suitable source, possibly tied to GPS signals. If is better than ∼1 part in 10ˆ9, then I think it's OK to ignore it, otherwise it should probably be investigated for its impact on wind accuracy. Either way, it should probably be discussed. Measuring lines on either side of the LO can significantly reduce sensitivity to that term (at the expense of wind precision), but if the measurement approach relies on that it should certainly be discussed.

Overall, the level of detail is well suited to the topic, and the standard of English and graphics is very good (though there are few points here and there I tried to capture in my comments below).

—- Specific comments

— Page 1

Line 2: "... platform, with a launch planned for near 2022. It is ..."

Line 6: "... perpendicular directions in order to reconstruct ..."

Line 7: Consider putting commas before and after "near 655 GHz"? Also add "amount

of" between "small" and "wind"

Line 10: First word "the" -> "a"

Line 17: ".... parameters and for study of methods ..."

— Page 2

Line 7: First word "of" -> "in"

Sentence spanning lines 7, 8, and 9: I'd turn this sentence around: "Some important species, such as HO2 and ClO, have their clearest signals in this region of the spectrum (refs.)" or something similar.

Line 11: "... and measurements are not perturbed by ..."

Line 16: "... have difficulties in reproducing it where ..."

Sentence spanning lines 20-22: Again, I'd turn this around: "As climate and weather models increase their vertical range to encompass more of the stratosphere and mesosphere, the need for measurements to improve the accuracy of models in this region, and hence at lower altitudes, can be expected to rise", or something like that.

— Page 3

Discussion in first paragraph: Would be good to mention the WINDII and HRDI instruments and UARS. Was the information they provided not useful for your purpose, or at least some aspects of your purpose? Even though it was a while ago, were there not some questions that those instruments answered?

— Page 4

Lines 5-10: If you're nodding the spacecraft, presumably the rotation axis of that nod is along the flight direction. Does that not give the two tangent points a non-vertical locus? Is the choice to alternate the two views between the up and down scans intended to make them more vertical? If so it would be good to mention that explicitly.

— Page 5

Table 1: 1 MHz for the spectrometer resolution seems a bit on the coarse side to me, given the upper stratosphere / mesosphere target. Have studies been performed to see if finer resolution (e.g., some "zoomed in" lower bandwidth spectrometers on selected lines) might not improve the wind measurements?

— Page 9

Line 24: If it's not too difficult, it would be nice to quantify "small" (e.g., of order 10 cm/s?)

— Page 11

Line 21: "AURA/MLS" -> "Aura MLS"

Line 27: You cite Figure 2, but that figure shows the coverage for the SIW orbit, not the Aura orbit. I don't see the need for a second figure, so perhaps it's simpler just to remove the citation of Figure 2 here?

— Page 12

Line 13: "that corresponds to" -> ", corresponding to"

— Page 13

Line 4: I completely understand your dropping the non-diagonal terms in Sy, but it seems a shame after you went to such lengths to compute them long hand. Given the power of computers these days, is it still too much work to compute the full matrix inverse, at least once, and see what difference it makes? I guess it is rather large, so probably not. In which case, why to go such lengths to take up space in the earlier sections defining it? It might simply be easier to tell us up front that you plan to ignore those terms and explain why that's OK, rather than exposing the reader to the full algebra only to discard it.

[Figure]

— Page 17

Line 18: "First, we note that, except for O3 and H2O, all ..."

Line 30: Actually, ClO can be non-zero at night in some cases.

— Page 18

Line 24: Comma needed after "On the other hand"

---

## Author Comment (AC1) · 10 Jul 2018

We would like to thank the anonymous reviewer for his comments. We believe they help us to improve the manuscript and give us some matters to improve the mission. Here below are our answers. A corrected version of the paper has been uploaded as a supplement.

Please note that we found an error in Table 1. The IF range is 10.975-18.975 GHz and not 10.075-18.075 GHz.

Reviewer comments are in blue, the citations from the original manuscript are in italic and the manuscript modifications are in red. The answer "Done" simply means that the manuscript has been modified following exactly the reviewer comment.

1) Although primarily focusing on wind retrievals the manuscript does not address the question of local oscillator stability. However, for a wind error below 1 m/s an oscillator stability of df/f<3.3e-9 is needed. As the operation geometry does not allow observations in opposite viewing directions as done by other wind measurement techniques this stability needs to be long-term (i.e. for the entire mission lifetime) in order not to introduce trend artifacts. What are the expected frequency stability during heating/cooling of the satellite? What is the aging of the planned frequency source? Are there plans for a method to monitor the LO frequency? Please comment on this issue in the manuscript. Even if very high stability could be achieved and and the induced bias was marginal, an upper limit should be indicated. I assume it is one of the questions a reader has when reading about Doppler shift measurements.

The LO stability is indeed a key parameter when measuring the winds. We should have discussed this issue in the paper and we have included a discussion in Pages 13-14 based on the instrument design described in the proposal.

Here are some additional information not included in the corrected manuscript:
In the instrument design report, it is stated that "a highly stable 10 MHz TCXO (Rakon RPT7050) is used as frequency reference to the PLL circuit. This TCXO has a long term stability (aging) of only ±1 ppm/year (>0.5 MHz). The VCO used is HMC529LP5 from Analog Devices. The output frequency of the LO unit is 13.293 GHz."
A short term (24 hours) LO stability of 2 kHz (df/f = 3*10^-9) has been required, it corresponds to a wind error of 1 m/s. The instrument team considers that such a performance is challenging and they guaranty a stability of 10 kHz. However discussions with the instrument team are still going on and we expect to improve the performances for both long and short timescales using different hardware or connecting SIW to the Innosat Spacecraft bus clock.

Table 2 (P13) has been modified to include short-term (24 hours) and long-term (1 month) local oscillator frequency variability: 2-10 kHz and >0.5 MHz, respectively.

Page 14, Line 4. The paragraph has been rewritten in order to include a discussion on the LO frequency uncertainty:

*Systematic retrieval errors emerge from uncertainties on the instrument, calibration and forward model parameters, and LOS angles (Tab. 2).*
*It is difficult at this stage of the mission definition to provide proper* values for these uncertainties*. The* given *values are relatively close to those expected but rounded in the way that it will be straight-forward to linearly scale the retrieval errors according to any future better knowledge of the parameter*

*uncertainties. One may notice that the uncertainty on the line broadening parameter (Gi) is likely underestimated and the actual values should be between 1–4 % depending on the line. On the other hand, the calibration parameters are likely overestimated. Anyway these errors induce a relatively constant retrieval bias that could be mitigated with ad-hoc corrections if their properties are well understood, e.g., time scale and latitudinal variabilities* (see for example the JEM/SMILES data analysis in Baron et al. (2013b)).

Given the proposed design of the instrument (Murtagh, 2016), the 24-hours variability of the local oscillator frequency is between 2 and 10 kHz which straightly results to a LOS wind retrieval uncertainty of 1–5 m/s. The lower limit corresponds to the scientific requirement and the upper one is the worse acceptable case. Though it is a systematic error, it changes from one scan to another with a time correlation that has to be determined before launch. The impacts on other retrieved parameters are negligible. The 1-year frequency variability may be relatively large (>0.5 MHz or 250 m/s) and we should consider that an absolute frequency knowledge, good enough for retrieving winds, may not be available. The frequency calibration will be performed using short-term wind retrieval bias estimates within 40–60 km where other systematic errors are small.
Retrieval errors from other parameters are investigated in Sect. 5.2 using a perturbation method:
*EQ17 ...*

And in the conclusion (P22,L5):
Hence ad-hoc methods for reducing retrieval biases must be studied. These methods can be used to calibrate the LO frequency long-term trend that may arise with the proposed hardware. However, improvements of the instrument design for following the frequency trend with a precision better than 2 kHz, are still being investigated.

Methods for mitigating wind retrieval bias have to be defined. Looking at opposite directions such as it is done with a ground based instrument, is difficult in space (e.g., solar illumination issue) and not be efficient (the two opposite measurements will be 2000-3000 km away from each other). Other methods more likely based on daily zonal statistics have to defined. For instance we may use the fact that systematic errors lead to zonal-wind retrieval errors with opposite sign on the ascending and descending orbit branches.
This issue has been added in Appendix A:
...
A systematic error $e\_los$ on the LOS wind retrievals propagates to the *U* and *V* components as follows:
1. The systematic error on the zonal-wind estimate is $e\_u=e\_los\ (cos(alpha\_n) + sin(alpha\_n))$
2. The systematic error on the meridional-wind estimate is $e\_v=e\_los\ (cos(alpha\_n)-sin(alpha\_n))$

We assume that $e\_los$ does not depends on the LOS orientation which is a valid assumption for the errors investigated in this paper (LO frequency, calibration, spectroscopy). We should note that $e\_v = 0$ for *phi\_n* = 45 deg or 225 deg which occur at latitudes between 30N–50N on the ascending branch of the orbit and between 10N–30N on the descending branch. The cases $e\_u = 0$ occur for the retrievals at the lowest and highest latitudes.
At the equator, the bias on the meridional wind is partly canceled out and the bias correction method used for JEM/SMILES analysis may not be satisfactory. For instance, an error $e\_los = 1$ m/s  induces an error $e\_v=0.2$ m/s.  On the other hand, the error on the zonal component is 1.4 m/s with an opposite sign on the ascending and descending orbit branches. The sign difference may provide us with a way to characterize LOS wind retrieval systematic errors.

2) You suggest a sun-synchronous orbit crossing the equatorial ascending node at 18:00. I agree that this allows perfect conditions for wind observations by allowing to constantly observe the night side with the higher ozone concentrations. My concern about this choice is that the representativeness of the measurements of trace gases near the day-night terminator may be delicate as during this period the concentrations of photochemically active species undergo rapid changes. This may among others introduce an articial annual cycle to your measurements simply by modifying the time before (after) the sunrise and after (before) the sunset the observations are made. Can you quantify this effect and what are your ideas to mitigate it?

Unfortunately the choice of the LT ascending node is fixed by the launcher and, in order to keep low Innosat mission budget, the choice is limited. This being said, as explained by the reviewer, wind and temperature retrieval performances are better in nighttime conditions but reactive species should be measured both in day and night times as for Aura MLS or MIPAS.
There is a compromise to be done. For this mission, we have chosen the most favorable conditions for wind measurements which are very challenging. Also, flying near the terminator provides favorable conditions in term thermal stability and solar-panel illumination.

The analysis of the species with diurnal variations (meso-O3, ClO, HO2, NO) will be performed with photo-chemical models. We will benefit from all the studies and methods implemented for previous missions like Odin or ACE/FTS. Most of the observations will be performed at SZA (+/- 10-20 deg from the terminator) when abundance changes will be slower than at sunrise or sunset.

*Can you quantify this effect and what are your ideas to mitigate it?* This is a very broad topic that cannot be addressed here. Each molecule, altitude, latitude and season should be treated as a special case.

**Specific comments**

**Page 1, L1: Why is your instrument called \Stratospheric Inferred Winds" when effectively assessing the upper stratosphere and lower mesosphere?**

Siw is a chacracter of the Swedish mythology. The acronym put the focus on the stratosphere and wind. Though winds are only measured over about half of the stratosphere (30–50 km), the mission is primary a stratospheric mission. Middle and upper stratospheric winds are a key product of the mission as explained in the paper. The mission is also able to provide a rather comprehensive description of the full stratosphere including high precision temperature and O3 measurements as well as good measurements of important species for studying its chemistry, dynamics and radiative budget (H2O, HCl, N2O, HNO3, ClO or HCN).

Good measurements of the mesosphere will also be performed but the instrument design is not optimized for such altitudes. A spectral resolution of 0.5 MHz would have been better as well as having an additional spectral channel with a strong line for improving the temperature retrieval performances, temperature being a key parameter for inverting molecular lines.

p.2, l.5: "risk of an observation gap in the near future" Is there no citation for this statement?

We are not aware of a refereed paper discussing this issue. However the following presentation at a recent SPARC meeting is available on the Web. We have added its reference in the manuscript.

Livesey, N. J. and Santee, M. L.: Prospects for future spaceborne measurements of interest to the SPARC DA Community and how to improve those prospects, in: S-RIP 2017 and 13th SPARC-DA Workshop, https://events.oma.be/indico/event/18/material/slides/16.pdf, 2017.

p.2, l.20: You could maybe add a citation of a modeler stating that wind simulations in these regions are hard to obtain.

The references given in Line 20 (Baron et  2013, Pichon et al., 2015….), though they focus on the measurements, clearly shows difficulties of analysis and re-analyses to reproduce wind in the mesosphere. Because of the lack of measurements, most of the GCM wind evaluations have been performed for altitudes below 10~hPa.  For instance, the difficulties for a GCM to reproduce Equatorial wind at 10 hPa has recently been discussed in Kawatani et al., (2016).  A key dynamical feature of the upper part of the  middle-atmosphere is the vertically propagating tides. Using temperature data, Sakazaki et al. (2018) found significant tide signature differences between the latest re-analyses and measurements of in the upper stratosphere and lower mesosphere. We have added these references as:

*Modeling middle-atmospheric major dynamical phenomena such as* vertically propagating tidal waves*, high-latitude sudden stratospheric warming or equatorial quasi-biennial oscillation are still challenging (Limpasuvan et al., 2012; Newman et al., 2016; Orsolini et al., 2017;* Sakazaki et al., 2018*). Wind … cannot be described by the geostrophic approximation such as in the equatorial region where the Coriolis force is weak and, in the upper stratosphere and mesosphere where wave*s *and tides*  *tend to dominate the wind fields (Baron, 2013, LePichon, 2015,* Kawatani et al., 2016*….*

Kawatani, Y., Hamilton, K., Miyazaki, K., Fujiwara, M., and Anstey, J. A.: Representation of the tropical stratospheric zonal wind in global atmospheric reanalyses, Atmospheric Chemistry and Physics, 16, 6681–6699, doi:10.5194/acp-16-6681-2016, https://www.atmos-chem-phys.net/16/6681/2016/, 2016.

Sakazaki, T., Fujiwara, M., and Shiotani, M.: Representation of solar tides in the stratosphere and lower mesosphere in state-of-the-art reanalyses and in satellite observations, Atmospheric Chemistry and Physics, 18, 1437–1456, doi:10.5194/acp-18-1437-2018, https://www.atmos-chem-phys.net/18/1437/2018/, 2018.

p.2, l.24: does -> do
Done
p.2, l.25/27: The measurement approach presented in Baumgarten 2010 is also providing wind in the gap region. Please add citation at l.27.
Done

p.2, l.30: Rather 20 or 30 km?
Using the Rayleigh channel (signal backscattered by molecules), LOS wind can be retrieved up to about ~30 km with a precision better than 5 m/s and a resolution of 2 km.

Do you see Aeolus as a good complement to your mission if you get synchronous mission activity? Please comment on this.
In case of overlap between both missions, there is an obvious complementary since Aeolus targets wind below 30 km and SIW targets higher  altitudes. However, in the altitude range between 20-35 km, the performances of both missions are weak. Also, Aeolus measures only the wind component along a single line-of-sight. These information are given in the paper and further discussions about the complementary of both missions are out of the scope of this paper.

Using Aeolus data together with SIW for scientific studies will be investigated  though Aeolus lifetime is officially between 2018-2021. More generally, combining lidar and micro-wave or infra-red passive sensors should be the solution in the future for measuring winds  from the surface to the lower thermosphere.

p.3, l.1: Please clarify that the wind profiles published in Wu et al. 2008 do not cover the gap region you defined as 30-70 km.

The sentence has been rephrased as follows:

 The potential of MM/SMM limb sounders for measuring winds has been demonstrated with LOS wind retrievals between 70–90 km from  MLS O2 line (Wu et al., 2008) and between 30–80 km from O3 and HCl lines measured with  Superconducting Submillimeter-Wave Limb-Emission Sounder (SMILES) (Baron et al., 2013b).

p.3, l.9: Please indicate the expected lifetime of SIW. Is there a chance to have it observing at the same time as SMILES-2? Would there be an added-value if both missions would be observing synchronously or would the expected higher performance of SMILES-2 make SIW obsolete?

Given the uncertainties on SMILES-2, having SIW is a very good thing. SIW lifetime is planned to be 2 years in order to allow the launch of a new Innosat mission every two years. If SMILES-2 is selected this year, the launch should be near 2025. In this case, SMILES-2 will follow SIW without overlaping time.

However, beside the budget consideration, SIW hardware lifetime could be longer than 10 years (e.g., as for MLS and SMR instruments) and a time expansion of the mission may then be possible. Having an overlap with SMILES-2 will benefit to both missions, and the quality of the scientific outcomes will be improved. For instance, the spatial and local time samplings are different and complementary. The fixed local times of SIW measurements will help to characterize non-diurnal changes in the SMILES-2 dataset such as non-migrating tide effects on temperature and wind. A long-term database can be produced with SIW that can not be done with SMILES2 whose the lifetime will be shorter than 5 years due to the limited lifetime of the cryo-cooler.
In term of data processing, we will share problems and information related to the 655 GHz band that has never been measured before (spectroscopic data, retrieval strategy, ...). Collaborations between both teams already exist and they will be strengthened if SMILES-2 is launched. This will lead to more efficiencies for defining, implementing and validating the processing chains.

The sentence p.3, l.9  has been rephrased as follows:
… a*nd SIW has been selected for the 2nd launch near 2022*. It will  observe the middle-atmosphere (15–90 km) for a period expected to be at least 2 years, and will provide *horizontal-wind vector within 30–90 km.*

Sect. 2.1: Please extend the instrument description. It is clear that this is not an instrument paper, but some core characteristics of the receiver should be introduced here. This will moreover avoid that questions arise during the further read of the manuscript.

In this is manuscript, we want to focus only on the main instrument characteristics that are relevant for the simulations. We do not want to go too deep in the description of hardware details. Moreover some

of them may still be modified for optimizing the performances. We do not wish to add more details but, if the reviewer believes that an important information is missing, we will be happy to add it.

P.4, l.6: With your scanning scheme LOS winds at 45 and 135 deg will be recorded from a similar location only for 1 altitude of your scan. How large will the distance between this two components be at maximum? Is this sampling mismatch not critical for your calculations of one zonal and meridional wind profile with Eq. (A2)?

The maximum distance between 2 scans is less than 400 km. This is equivalent to the LOS horizontal resolution. No significant errors should be induced by the position mismatch.
The manuscript has been changed has follows (the modifications also includes reviewer 2 comment answer):

The forward antenna is used during the upward scans and the aftward one during the downward scans. With this choice, the horizontal displacement of the tangent point during a vertical scan is less than 300 km, the vertical motion of the line-of-sight partly counterbalancing the satellite motion. Using the line-of-sight (LOS) winds retrieved with the two antennas over close regions allows us to derive the meridional and zonal wind components (Appendix A). The separation between the LOS wind profiles is less than 400 km.

p.4, l.7: "continuously rotate" is misleading as it is in fact not an unaccelerated rotation but rather a succession of upward and downward scans.
I think this was an issue corrected after the quick review process. In the current version, the text is:
"… *the whole satellite will nod up and down in order to scan the limb alternatively upward and downward from about 15 to 90 km*"

p.4, l.28: "at least a factor of 2... compared to other spectral regions." Please be more concrete. What are "other spectral regions".
The sentence has been rephrased as follows:
*… a factor 2 the wind measurement sensitivity between 40–70 km compared*  to retrievals performed from a band with similar characteristics but located at any other frequency under 800 GHz.

Fig. 3: You display 9 GHz but the spectrometer bandwidth is 8 GHz. Please mark the (un)used frequency range in this figure.
The ranges outside the spectral bandwidth are now indicated with grey-shaded areas. The caption has been updated accordingly.

Fig. 3: What is the reason for the 2 GHz frequency shift compared to SMILES-2 (Ochiai et al 2017)?
There is no frequency difference with the band shown in Ochiai et al. (2017). In Ochiai et al., the band is displayed differently: each sideband is divided in two ranges of 4 GHz.

Fig. 3: Displaying the centre frequecy of LSB and USB directly in the different panels would help to further clarify the figure.
The central frequency is now indicated in the plot x-labels.

p.5, l.7: I suggest to modify "so-called brightness temperature" to "so-called Rayleigh-Jeans brightness temperature" to make sure the user is not confused by the 1 mK of cosmic background (as I was at first).

Done

Done

Fig. 4: It is not completely clear from the text what you intend to communicate to the reader with this figure. Please extend the description and reasoning in the text.
The text P6 l.7 is changed as follows:
*and I is the specific intensity*. The specific intensity is integrated along a LOS as that shown in Fig. 4. The LOS is characterized by the altitude of the tangent point (i=0), the angle with the meridional direction *phi_n* and narrow ranges *i* over which the atmosphere is considered homogeneous.

The index "*i*" (frequency) in Eq (3) is replaced with "k" to avoid confusion with the LOS range index used in the figure.

Eq. (2): Why do you use quadratic addition of the static antenna pattern and the broadening due to the scanning velocity? I would argue that, if you combine the static pattern with the scanning, your beam pattern becomes non-Gaussian. In any case, I think that by quadratic addition you drastically underestimate the width of your main lobe unless the scanning broadening is much smaller than the static beam width. If not the case, I would think that a linear addition would already be closer to reality while still underestimating the width of your beam (due to the non-Gaussicity introduced by the scanning). Please review the information about the beam width in the manuscript.

We agree with the reviewer that considering the effective antenna pattern as a Gaussian is an approximation. But it is a satisfactory approximation for a sensitivity study. The static antenna vertical width is about 5 km which is significantly larger than the altitude range scanned during the spectrum integration (1.1 km). Moreover the retrieval vertical resolution is 5 km, so such an approximation has no impacts on the results. Antenna side-lobes have also small impacts on the retrieval errors estimation and can be neglected.
The text is changed in order to clarify these points:

Given that the altitude range scanned during the spectrum integration is small compared to the static antenna vertical resolution (1.1 km and 5 km, respectively), the effective antenna pattern including the vertical scan, *is approximated by a Gaussian function …*
The antenna sidelobes are also neglected. These approximations have negligible impacts on this study.

p.8, l.14: Why are you using JPL for some lines and Hitran for others?

The JPL lines used in this work are not in HITRAN. The text is changed as follows:
… that are not available in HITRAN and are, then, taken from the Jet Propulsion Laboratory catalog (Pickett et al., 2018).

p.8, l.16/17: You could indicate worst-case bias induced by these effects to show how marginal they are.
There is a lack of information for the pressure shift parameters. In HITRAN 2016, only values for HCl and H2O lines are available. At 10 hPa (~30 km), the H2O line shift is 230 kHz and an error of 2% corresponds to a wind error of ~2 m/s. This value should be smaller for O3 lines.

The text is changed as follows:

*The line frequency is also shifted by pressure but this effect is small above 25 km where winds are measured.* For the H2O line at 620 GHz, 2% error on the shift parameter corresponds to an error of 2 m/s at 10 hPa. The shift on O3 lines should be smaller but the information is not available in HITRAN and further studies are needed to infer it.

p.10, l.1: You refer to Tso as antenna spillover. Looking at Eg (8) Tso rather refers to the average brightness temperature of the regions where the radiation which you receive because of the spillover in your optics actually comes from. Please adapt the wording.

The text has been rephrased as follows:

T_so is the mean brightness-temperature introduced by the optics spillover, ...

p.10, l.2/l.15: "spill-over" or "spillover". Please use consistent spelling

The term "spillover" is now used instead of "spill-over"

Eq. (10): You may state that you assume Tc = 0 here.

We changed the text as follows:

Assuming a linear response of the radiometer and using T_c << T_h, the radiometer gain is derived

Eq. (15): Does this linear approach suffice for all situation you expect to encounter? What happens if the truth is further away from the first guess than in Fig. 6? Is this linear retrieval also sensible for photochemically active species close to the day/night terminator?

The retrieval approach presented here is good enough for estimating the retrieval errors with respect to the atmospheric state. This is a very common approach used for other mission studies such as MLS, Odin and SMILES. The definition of a robust retrieval algorithm is not needed and it is not discussed in this paper. Let's note that methods to handle non-linear effects that can arise for cases with large differences between the first guess and the true atmosphere, exist. For instance, the linear scheme presented in the paper can be integrated into a standard Levenberg-Marquardt iterative scheme (e.g., see Urban et al. 2004 and Baron et al. 2011 given in the paper). For species such as ClO, N2O, HOCl the problem is nearly linear even for large differences and a linear approach should provide good performances even near the terminator. The main issue in this case is possible large horizontal inhomogeneities near the tangent point. But such cases will be rare (see 2[nd] main comment answer).

Eq. (15): You use Sdy instead of Sy. What about error correlations? Can they be neglected and why? Please state it in text.

The correlations are not neglected. They are taken into account in Eq. 16 since the full matrix Sy is used. The diagonal matrix *Sdy* is only used in the inversion of K as a measurement weight.

The reason to not invert the full matrix is because it is too large (~8000 frequencies and 150 tangent heights) to be done with the computer used for this analysis. In the future, we will optimize the computations in order to reduce the matrix size and use sparse matrix algorithms for the inversion. In theory (i.e, if the frequency correlations are properly characterized), the retrieval errors will slightly be decreased.

We changed the text as follow to make it clearer:

P13L10: "… *the standard deviation of x,* Sy is the full measurement error covariance matrix (Eq. 14) *and ...*"

p.13, l.15: Please add the reason for the increasing error at lower altitudes here.

*lower altitude for wind retrieval.* The error increase is due to the pressure broadening of the lines that is about 20–40 MHz at 10 hPa.

Done (as well as in Table 2 and A1  captions)

The elevation offset is one of the retrieval parameter in x. This is explained at the beginning of the section (P12,l10):
*"The retrieved state x_hat is a vector including all the unknown parameters of the forward model, namely the atmospheric vertical profiles, a radiance offset on each spectrum and a mean pointing angle offset of the whole scan."*

Since it is a  standard approach, we do not think that we need to provide more details.

p.16, l.10: Please indicate the reason why the best performance is found over the northern polar regions.

The mesospheric wind measurement performances are the best over the night-time poles because of the O3 enhancement. The text is changed as follows:
*"The best performances are found over the northern polar region where* the nighttime \chem{O_3} enhancement is the largest. There, the *LOS-wind can be retrieved with a precision better than..."*

Sect. 5.2: Why using a 10 times to large error for the sideband ratio? Indeed a 1% sideband uncertainty seems rather large. Please consider to modify it to the value of 0.1% that you found in your preliminary study. The choice of 1% uncertainty which you then qualify several times as too large unnecessarily complicates the reading of this section.

We changed the manuscript to consider an error of 0.1% on the DSB parameter. The x-scales of Fig. 11 and 12  have been changed accordingly as well as the DSB error discussions in sections 5.2.1 5.2.2 and conclusion (P22,L1).

Done

Done

Done

The sentense has been rephrased as follows:
*"The retrieval of  two line-of-sight winds over the same region allows us to compute … "*

Done

Eq. (B1): What is the significance of "max" here? eps_(x,M) seems to be a scalar (see Eq. (18)) so I don't see what you want to do by taking the maximum.

This term eps_(x,M) means the set of the errors induced by all the parameters of all the lines of a given species M. In the revised version, it is replaced by *{eps_(x,M,pi)}_pi* and the text is rephrased as:

"where M denotes the chemical species, *eps_(x,M,pi)* is the error induced by the parameter *p* of the line *i* (Eq. 18), and *{eps_(x,M, pi)}_pi* is the set of errors induced by all the parameters of all the lines of the species M."

---

## Author Comment (AC2) · 10 Jul 2018

We would like to thank the anonymous reviewer for his comments. We believe they help us to improve the manuscript and give us some matters to improve the mission. Here below are our answers. A corrected version of the paper has been uploaded as a supplement.

Please note that we found an error in Table 1. The IF range is 10.975-18.975 GHz and not 10.075-18.075 GHz.

The reviewer comments are in blue, the citations from the original manuscript are in italic and the manuscript modifications are in red. The answer "Done" simply means that the manuscript has been modified following exactly the reviewer comment.

1) I think the authors have somewhat glossed over the trade inherent in such measurements between precision and resolution (vertical resolution mainly in this case). They can afford not to dwell on it in this study because they've chosen a rather coarse 5km spacing for their state vector resolution, but I do think it deserves some mention (but probably not any new calculations). At no point do they discuss averaging kernels, but I imagine they've computed them, or could do so very easily. I expect that such kernels indicate that the information content is a good match for their 5 km grid (i.e., that A is a good approximation to the identity matrix, except perhaps at the top and the bottom of the profiles). I would however like them to add a discussion to that effect somewhere in the text. Their mention of the potential of finer 3km resolution for temperature (page 17, line 7) implies that they have looked into this issue somewhat. I encourage them to talk about it just a bit more.

A weak regularization is used for inverting the forward model Jacobian matrix which gives unity averaging kernels. The vertical resolution is then given by the resolution of the retrieval altitude grid. We have chosen to use the same resolution of 5 km for every product over the whole altitude range in order to simplify the discussion. The resolution corresponds to the antenna field of view resolution (i.e., the vertical resolution of the measured radiance). However proper retrieval algorithms will consider the best resolution for each product considering the trade-off between vertical resolution and precision. The manuscript has been modified to clarify these points as follows:

Page 12, Line 12:
"The retrieval altitudes  range … *The  grid vertical resolution is 5~\unit{km} that corresponds to the effective* vertical *field-of-view of the instrument*"

Page 13, Line 4:
"*… Ux is a diagonal matrix for stabilizing the matrix inversion. Its element square-roots correspond to very large standard deviations* (STD) *of x, typically > 10000%, 1000 K, 1000 m/s for VMR, temperature and LOS wind, respectively. The regularization effects are negligible where*  the retrieval errors (null space and measurement errors)  are much smaller than the *Ux* related STD. In other words, the averaging kernels are unity at altitudes where the measurement is relevant and the retrieval vertical resolution is that of the retrieval altitude grid. *"

Page 15, Line 4:
 *The precision (1-$\sigma$) is given for ~~a retrieval vertical resolution of 5~\unit{km} andand for a single scan~~. It is possible to use the altitude information inscribes in the pressure broadened lineshape, for retrieving atmospheric profile with a better resolution but at the cost of the precision. Precision degradation can be afford for products retrieved from strong signals  (e.g., \chem{O_3} or temperature) or for those whose the vertical resolution is more scientifically relevant than the temporal or horizontal one (precision can be

improved by averaging data). On the other hand, degrading the vertical resolution may be necessary for providing useful information on products derived from weak signals (e.g, \chem{HOCl}). Later, using the results of this study and based on scientific requirements, the retrieval algorithm will be optimized for providing the best compromise between precision and resolution for each of the main products. …*Also the retrieval vertical resolution can be increased for improving the precision of species with weak lines.*

2) The authors make no mention of frequency stability requirements for the instrument Local Oscillator (presumably tied to some lower frequency clock source). I presume the instrument (or spacecraft) design includes some suitable source, possibly tied to GPS signals. If is better than 1 part in 10ˆ9, then I think it's OK to ignore it, otherwise it should probably be investigated for its impact on wind accuracy. Either way, it should probably be discussed. Measuring lines on either side of the LO can significantly reduce sensitivity to that term (at the expense of wind precision), but if the measurement approach relies on that it should certainly be discussed.

The LO stability is indeed a key parameter when measuring the winds. We should have discussed this issue in the paper and we have included a discussion in Pages 13-14 based on the instrument design described in the proposal.
Here are some additional information not included in the corrected manuscript:
In the instrument design report, it is stated that "a highly stable 10 MHz TCXO (Rakon RPT7050) is used as frequency reference to the PLL circuit. This TCXO has a long term stability (aging) of only ±1 ppm/year (>0.5 MHz). The VCO used is HMC529LP5 from Analog Devices. The output frequency of the LO unit is 13.293 GHz."
A short term (24 hours) LO stability of 2 kHz (df/f = $3*10^{-9}$) has been required, it corresponds to a wind error of 1 m/s. The instrument team considers that such a performance is challenging and they guaranty a stability of 10 kHz. However discussions with the instrument team are still going on and we expect to improve the performances for both long and short timescales using different hardware or connecting SIW to the Innosat Spacecraft bus clock.

Table 2 (P13) has been modified to include short-term (24 hours) and long-term (1 year) local oscillator frequency variability: 2-10 kHz and >0.5 MHz, respectively.

Page 14, Line 4. The paragraph has been rewritten in order to include a discussion on the LO frequency uncertainty:
*Systematic retrieval errors emerge from uncertainties on the instrument, calibration and forward model parameters, and LOS angles (Tab. 2). The most critical parameters are investigated using a perturbation method:*
*It is difficult at this stage of the mission definition to provide proper* values for these uncertainties. *The* given *values are relatively close to those expected but rounded in the way that it will be straight-forward to linearly scale the retrieval errors according to any future better knowledge of the parameter uncertainties. One may notice that the uncertainty on the line broadening parameter (Gi) is likely underestimated and the actual values should be between 1–4 % depending on the line. On the other hand, the calibration parameters* *and sideband ratio* *are likely overestimated. Anyway these errors induce a relatively constant retrieval bias that could be mitigated with ad-hoc corrections if their properties are well understood, e.g., time scale and latitudinal variabilities* (see for example the JEM/SMILES data analysis in Baron et al. (2013b)).

The 24-hours variability of the local oscillator frequency is between 2 and 10 kHz which straightly results to a LOS wind retrieval uncertainty of 1–5 m/s. The lower limit corresponds to the scientific

requirement and the upper one is the worse acceptable case. Though it is a systematic error, it changes from one scan to another with a time correlation that has to be determined before launch. The impacts on other retrieved parameters are negligible. The 1-year frequency variability may be relatively large (>0.5 MHz or 250 m/s) and we should consider that an absolute frequency knowledge, good enough for retrieving winds, may not be available. The frequency calibration will be performed using short-term wind retrieval bias estimates within 40–60 km where other systematic errors are small.
Retrieval errors from other parameters are investigated in Sect. 5.2 using a perturbation method:
*EQ17 ...*

And in the conclusion (P22,L5):
Hence ad-hoc methods for reducing retrieval biases must be studied. These methods can be used to calibrate the LO frequency long-term trend that may arise with the proposed hardware. However, improvements of the instrument design for following the frequency trend with a precision better than 2 kHz, are still being investigated.

**Specific comments**

**Page 1**
Line 2: "... platform, with a launch planned for near 2022. It is …"
Done

Line 6: "... perpendicular directions in order to reconstruct …"
Done

Line 7: Consider putting commas before and after "near 655 GHz"? Also add "amount" of" between "small" and "wind"
Done

Line 10: First word "the" -> "a"
Done

Line 17: ".... parameters and for study of methods …"
Done

**Page 2**

Line 7: First word "of" -> "in"
Done

Sentence spanning lines 7, 8, and 9: I'd turn this sentence around: "Some important species, such as $HO_2$ and ClO, have their clearest signals in this region of the spectrum (refs.)" or something similar.
Done

Line 11: "... and measurements are not perturbed by ..."
Done

Line 16: "... have difficulties in reproducing it where ..."
Done

Done

**Page 3**

Discussion in first paragraph: Would be good to mention the WINDII and HRDI instruments and UARS. Was the information they provided not useful for your purpose, or at least some aspects of your purpose? Even though it was a while ago, were there not some questions that those instruments answered?

Both HDRI and WINDII are described in Shepherd et al. (2015). However it is true that HRDI should be explicitly cited since it is the only spaceborne instrument designed measuring wind in the stratosphere and mesosphere (WINDII measured winds above 90 km). The manuscript has been changed has follow:

Page 2, line 21-22:
 Only High Resolution Doppler Imager (HRDI) on Upper Atmospheric Research Satellite (1993-2005) has been able to measure horizontal winds over the stratosphere and mesosphere (Ortland et al., 1996), and current spaceborne sensors are not able to measure wind accurately below 90 km (Shepherd, 2015).

**Page 4**

Lines 5-10: If you're nodding the spacecraft, presumably the rotation axis of that nod is along the flight direction. Does that not give the two tangent points a non-vertical locus?

Is the choice to alternate the two views between the up and down scans intended to make them more vertical? If so it would be good to mention that explicitly.
The tangent height foot-print is not vertical. It moves along the orbit track with the satellite (7 km/s) and toward the satellite along the LOS in the ascending scan and away from the satellite during the descending scan. The nodding movement is performed over a small angle range, the deviation of the tangent points wrt to the vertical is small.

The text (P4,L6) has been modified as follow:
*… upward and downward from about 15 to 90 km.* The forward antenna is used during the upward scans and the aftward one during the downward scans. With this choice, the horizontal displacement of the tangent point during a vertical scan is less than 300 km, the vertical motion of the line-of-sight partly counterbalancing the satellite motion. *Using the line-of-sight* (LOS) *winds retrieved with the two antennas over close regions*  *allows us to derive the meridional and zonal wind components (Appendix A).* The separation between the LOS wind profiles is less than 400 km.

P4,L10:
The following sentence is removed:
*"The forward antenna is used during the upward scans and the aftward one during the downward scans."*

**Page 5**

Table 1: 1 MHz for the spectrometer resolution seems a bit on the coarse side to me, given the upper stratosphere / mesosphere target. Have studies been performed to see if finer resolution (e.g., some "zoomed in" lower bandwidth spectrometers on selected lines) might not improve the wind measurements?

The paper shows simulations based on the proposal status but the instrument design optimizations are still investigated such as the possibility to have different frequency resolution. A spectral resolution of 0.5 MHz for key mesospheric lines would improve the wind retrieval precision by more than 20% above 70 km (e.g., Fig. 6 in Baron (2013a)). Temperature retrieval should also be improved but further simulations have to be conducted to quantitatively assess the improvements.

The 200 MHz frequency range between IF = 17.2 and 17.4 GHz (LO=638.075 GHz) is the range that should be selected in priority. It contains the strong $H_2O$ line (620.701 GHz) and the two strongest $O_3$ lines (620.825 and 655.289 GHz).
The second priority would be to increase of the resolution for the NO lines (651.1 GHz).
In order to compensate the telemetry data increase, the resolution could be decreased in other frequency range such as in the spectral window near the $N_2O$ stratospheric line (652.8 GHz).

and we will not discuss this point in the main text. However this potential improvement is added in the conclusion:
P22L18: *"...could improve the measurement sensitivity by more than 20 %. Retrievals could also be improved in the mesosphere by increasing the frequency resolution to 0.5 MHz between the intermediate frequencies 17.2 and 17.4 GHz, a range that contains the strong $H_2O$ line (620.701 GHz) and the two strongest $O_3$ lines (620.825 and 655.289 GHz). Implementing such a setting is under investigation."*

**Page 9**

Line 24: If it's not too difficult, it would be nice to quantify "small" (e.g., of order 10 cm/s?)

The vertical width of the intensity weighting function is about 1 km. (WF are defined for a single ray before antenna convolution). At the equator, a horizontal wind of 100 m/s induces an error is 7 cm/s for a LOS point at 1km above the tangent point.
The error is given as (Ve + Vlos)*(1-sin(phi_i)) with Ve the Earth rotation speed along the LOS (<370 m/s) and Vlos is the horizontal LOS wind and phi_i the angle between the nadir direction and the LOS.

The manuscript is changed has follows:
P9, L24: *These errors are  smaller than 10 cm/s and have negligible impacts on the retrievals.*

**Page 11**

Line 21: "AURA/MLS" -> "Aura MLS"
Done as well as replacing AURA with Aura in other places.

Line 27: You cite Figure 2, but that figure shows the coverage for the SIW orbit, not the Aura orbit. I don't see the need for a second figure, so perhaps it's simpler just to remove the citation of Figure 2 here?

The black dashed lines show the Aura SZA vs Latitude for January.

**Page 12**

Line 13: "that corresponds to" -> ", corresponding to"
Done

**Page 13**

Line 4: I completely understand your dropping the non-diagonal terms in Sy, but it seems a shame after you went to such lengths to compute them long hand. Given the power of computers these days, is it still too much work to compute the full matrix inverse, at least once, and see what difference it makes? I guess it is rather large, so probably not. In which case, why to go such lengths to take up space in the earlier sections defining it? It might simply be easier to tell us up front that you plan to ignore those terms and explain why that's OK, rather than exposing the reader to the full algebra only to discard it.

The correlations are not neglected. They are taken into account in Eq. 16 since the full matrix Sy is used. The diagonal matrix *Sdy* is only used in the inversion of K as a measurement weight. The manuscript is changed to make this point clearer:
P13L10: "… *the standard deviation of x,* Sy is the full measurement error covariance matrix (Eq. 14) *and* ..."

The reason to not invert the full matrix is because it is too large (~8000 frequencies and 150 tangent heights) to be done with the computer used for this analysis.  In the future, we will optimize the computations in order to reduce the matrix size and use sparse matrix algorithms for the inversion. In theory (i.e, if the frequency correlations are properly characterized),  the retrieval errors will slightly be decreased.

**Page 17**

Line 18: "First, we note that, except for O3 and H2O, all ..."
Done

Line 30: Actually, ClO can be non-zero at night in some cases.
We agree and the statement has been softened:  "… *but vanish* in general *during nighttime.*"

**Page 18**
Line 24: Comma needed after "On the other hand"

Done